# Global comparative structural analysis of responses to protein phosphorylation

Miguel Correa Marrero [1,2,3] ✉, Victor Hugo Mello[4], Pablo Sartori [4] & Pedro Beltrao [1,2,3,4] ✉

Post-translational modifications (PTMs), particularly protein phosphorylation, are key regulators of cellular processes, impacting numerous aspects of protein activity. Despite widespread phosphorylation of eukaryotic proteomes, the function of most phosphosites remains unknown. Elucidating the structural mechanisms underlying phosphorylation is crucial for understanding its regulatory roles. Here, we present a comparative structural analysis of phosphorylated and non-phosphorylated proteins taken from the Protein Data Bank (PDB). Our study systematically evaluates how phosphorylation affects backbone conformation, protein dynamics, and mechanical strain. We found that phosphorylation commonly induces small, stabilizing conformational changes through conformational selection and frequently modulates local residue fluctuations, influencing overall protein motion. Notably, a small but significant subset of phosphosites shows mechanical coupling with functional sites, aligning with the domino model of allosteric signal transduction. This work provides a foundation for studying phosphorylation and other PTMs in their structural context, which will guide the rational design of synthetic phosphosites and enable the engineering of PTM-driven regulatory circuits in synthetic biology.

Cells need to sense intra- and extracellular conditions and adapt to changes in them. Post-translational modifications (PTMs) are a fast way to modulate protein activities in response to such changes at a small metabolic cost. Among the myriad types of PTMs, the best studied of them all is phosphorylation. This reversible modification involves the covalent attachment of a phosphate group to specific residues. In eukaryotes, this occurs canonically in serine, threonine, and tyrosine residues, and, in prokaryotes and archaea, histidine[1–3]. Its addition and removal are catalyzed, respectively, by hundreds of kinases and phosphatases[4,5]. Phosphorylation, through the addition of a double negative charge, can regulate multiple facets of protein function, including modifications to conformation, activity, localization, and interactions[6,7]. This remarkable versatility allows phosphorylation to play key regulatory roles in a plethora of biological processes, such as the cell cycle or apoptosis[8]. Additionally, dysregulated phosphorylation is linked to many diseases, including cancer[9,10], diabetes[11,12] and viral infections such as COVID-19 or HIV/AIDS[13,14]. Altogether, this makes protein phosphorylation a key regulator of protein function and cellular processes.

Fueled by major technological advances in mass spectrometry, the last two decades have borne witness to explosive growth in the number of identified PTMs[15,16]. In yeast and humans, high-throughput methods to identify phosphorylation sites in proteins (phosphosites) have shown that up to 75% of their proteomes are phosphorylated[17,18]. The bottleneck in the study of phosphorylation has now shifted from identification towards functional characterization: the function of ~95% of human phosphosites is unknown[8], and it is debatable whether all phosphosites are actually functional. It is thought that a fraction of detected phosphosites may simply be the product of off-target kinase activity, yielding phosphorylations with no discernible impact on

[1]European Bioinformatics Institute (EMBL-EBI), Wellcome Genome Campus, Hinxton, UK. [2]Institute of Molecular Systems Biology, Department of Biology, ETH Zurich, Zurich, Switzerland. [3]SIB Swiss Institute of Bioinformatics, Lausanne, Switzerland. [4]Gulbenkian Institute for Molecular Medicine, Lisbon, Portugal. ✉e-mail: correamarrero@imsb.biol.ethz.ch; beltrao@imsb.biol.ethz.ch

organism fitness[19,20]. This is consistent with the fast rate of divergence in phosphosites across species and with estimates that up to 65% of human phosphosites are weakly constrained across evolution[21–24].

In order to address this challenge of functional assignment, different computational and experimental approaches have been developed to detect and rank functionally important phosphosites. Such methods include the identification of highly conserved phosphosite hotspots across protein domains[25], applying machine learning to derive a phosphosite-functional score[26] or chemical genomics of phosphodeficient mutants[27]. While these methods can detect functionally important phosphosites, they inherently cannot identify the structural mechanisms by which phosphorylation controls protein function. This makes it challenging to understand the fundamental relationship between phosphorylation, a change in the structural properties of the protein and its biological function, or its lack thereof. A final obstacle is that phosphorylation, while it can act orthosterically (i.e., by direct phosphorylation of a functional site), often acts as an allosteric effector (i.e., it modulates regions of the protein distal to the phosphosite)[28]. Despite its biological importance, allosteric communication is still poorly understood[29,30], complicating the functional assignment of allosteric sites.

Most research on the effects of phosphorylation on protein structure has focused on the study of specific proteins, typically well-studied model systems such as kinases[31–36]. While these studies provide detailed mechanistic insight into the effects of phosphorylation on the proteins under study, there is a need to derive broader principles regarding how phosphorylation regulates protein structure and function. General studies have been difficult due to the historically limited number of phosphorylated structures available in the Protein Data Bank (PDB) and the difficulties associated to expressing proteins phosphorylated at specific positions until the development of genetically encoded phosphosites in the last decade[37,38]. Furthermore, amino acid substitutions intended to mimic a phosphorylated residue often fail to replicate the intended behaviour[39,40].

Some studies have approached these questions with a broader lens. Johnson and Lewis[41] conducted a manual comparative analysis of the phosphorylated and non-phosphorylated structures of 17 different proteins. They recognized the general importance of electrostatic effects, but noted a diversity of structural responses to phosphorylation. Xin and Radivojac[42] performed a PDB-wide analysis of the structural effects of multiple PTMs and found that phosphorylation produces conformational changes both at the global and local levels. However, these changes are often small, with only ~13% of phosphorylated proteins undergoing changes greater than 2 Å. They also found that phosphorylation tends to introduce hydrogen bonds and salt bridges in the local environment of the phosphosite. Recent studies have instead largely focused on integrating phosphoproteomics with protein structure data[43–46]. For example, Kamacioglu et al.[43] mapped mammalian phosphoproteomics data onto protein structures, categorizing phosphosites according to their physicochemical characteristics. They found that the predicted functional importance of phosphosites within the protein core depends on their potential to become exposed, thus highlighting the role of protein dynamics in phosphosite functionality. These studies underscore the need for more systematic approaches to link phosphorylation events with specific structural and functional outcomes.

In this study, we exploit the growing availability and redundancy of phosphorylated structures and their non-phosphorylated counterparts in the PDB to conduct a species-agnostic, systematic evaluation of the structural changes induced by phosphorylation. Because the PDB predominantly comprises structures resolved through X-ray crystallography, our analysis is largely focused on well-structured proteins. However, it is well-documented that the majority of known phosphosites are located in disordered regions, which often regulate biological processes through short linear motif-mediated interactions.

A conservative estimate places a lower bound of 15% of eukaryotic phosphosites located within structured Pfam domains, excluding those adjacent to domains, which would be expected to function similarly[25]. Our study concerns the mechanisms through which this fraction of phosphosites work; notably, phosphosites in structured domains are predicted to be more likely to be functional[26], highlighting the importance of studying their structural effects. Our goal is to uncover general principles that govern how phosphorylation regulates protein structure. We employ a three-pronged computational approach combining structural bioinformatics and biophysics methods, examining changes in structure, dynamics, and mechanical strain, and find multiple overall tendencies associated with phosphorylation.

## Results

We mined the PDB for proteins with both phosphorylated and non-phosphorylated solved structures in order to study changes taking place upon phosphorylation. Where available, we used multiple structures for each protein, whether phosphorylated or not, allowing us to take into account intrinsic structural variability. The dataset was filtered to ensure the quality of the structures, as well as consistency and sufficient coverage (see "Methods"). The curated dataset (Supplementary Data 1) contained 225 different proteins and 347 different phosphosites, 79% of which are single-site phosphorylations, a substantially larger sample size than a previous PDB-wide analysis (3.2× more proteins and 3.9× more phosphosites)[42]. Fig. 1a presents an overview of the collection and filtering steps, which includes steps to ensure structure quality, a minimum coverage of the full protein sequence and consistency (in terms of sequence overlap) of the structures. Supplementary Fig. 1 shows a summary of the filtered dataset. Supplementary Data 2 contains a summary of the different metrics obtained in this work for each phosphosite.

### Phosphorylation is associated to small, stabilizing conformational changes

We first evaluated the extent of global conformational change resulting from phosphorylation by quantifying backbone root mean squared deviations (RMSDs) between pairs of structures of the same protein. As expected, phosphorylation was significantly linked to global changes in backbone conformation (one-tailed Mann–Whitney $U$ tests; Fig. 1b). Nonetheless, most changes tend to be small (median backbone RMSD 1.14 ± 3.13 Å), with only 28.14% of phosphorylation events associated to changes ≥ 2 Å (Fig. 1b), slightly more than twice as previously reported on a smaller sample[42]. Moreover, we also found that the median RMSD among phosphorylated structures tends to be smaller than among their non-phosphorylated counterparts (one-tailed Wilcoxon signed-rank test, $T = 5993.5$, $p$-value = $4.9 \times 10^{-6}$) (Fig. 1b, c), indicating a tendency towards greater structural uniformity post-phosphorylation. We observed the same trends using a superposition-free structure similarity measure[47], showing the robustness of our results to the choice of metric (Supplementary Fig. 2). These results show that, overall, phosphorylation tends to lead to subtle changes that stabilize a particular backbone conformation, corroborating previous computational research[42,48] and nuclear magnetic resonance (NMR) studies of phosphorylated peptides[49,50].

Factors such as crystallization conditions or refinement methods can influence the outcomes of protein structure solution[51,52]. We asked whether observed similarities between protein structures may arise from batch effects due to preferred experimental conditions and methods in the same research group, and found that, while structures from the same research group tend to be more similar, this effect does not drive the observed trends (Supplementary Fig. 3). Altogether, this aligns with earlier findings that, while experimental conditions can influence observed structural differences between post-translationally modified and unmodified proteins, they are insufficient to explain such changes[42]. We also asked whether protein-protein interactions (PPI)

might contribute to the conformational changes seen upon phosphorylation. A robust linear model showed that although PPI have a small but statistically significant effect, they cannot meaningfully explain the backbone differences between phosphorylated and non-phosphorylated structures (pseudo-$R^2 = -0.028$) (Supplementary Fig. 5). This finding aligns with previous observations that ligand binding likewise fails to account for these structural differences,

suggesting that complex formation in general does not drive the conformational changes associated with phosphorylation[42].

Another potential source of bias is over-representation of certain protein families, as some are researched more extensively than others. We examined the distribution of phosphosites across protein families and found that 37% occur within protein kinase domains, far exceeding any other Pfam domain class (Supplementary Fig. 4a). When we

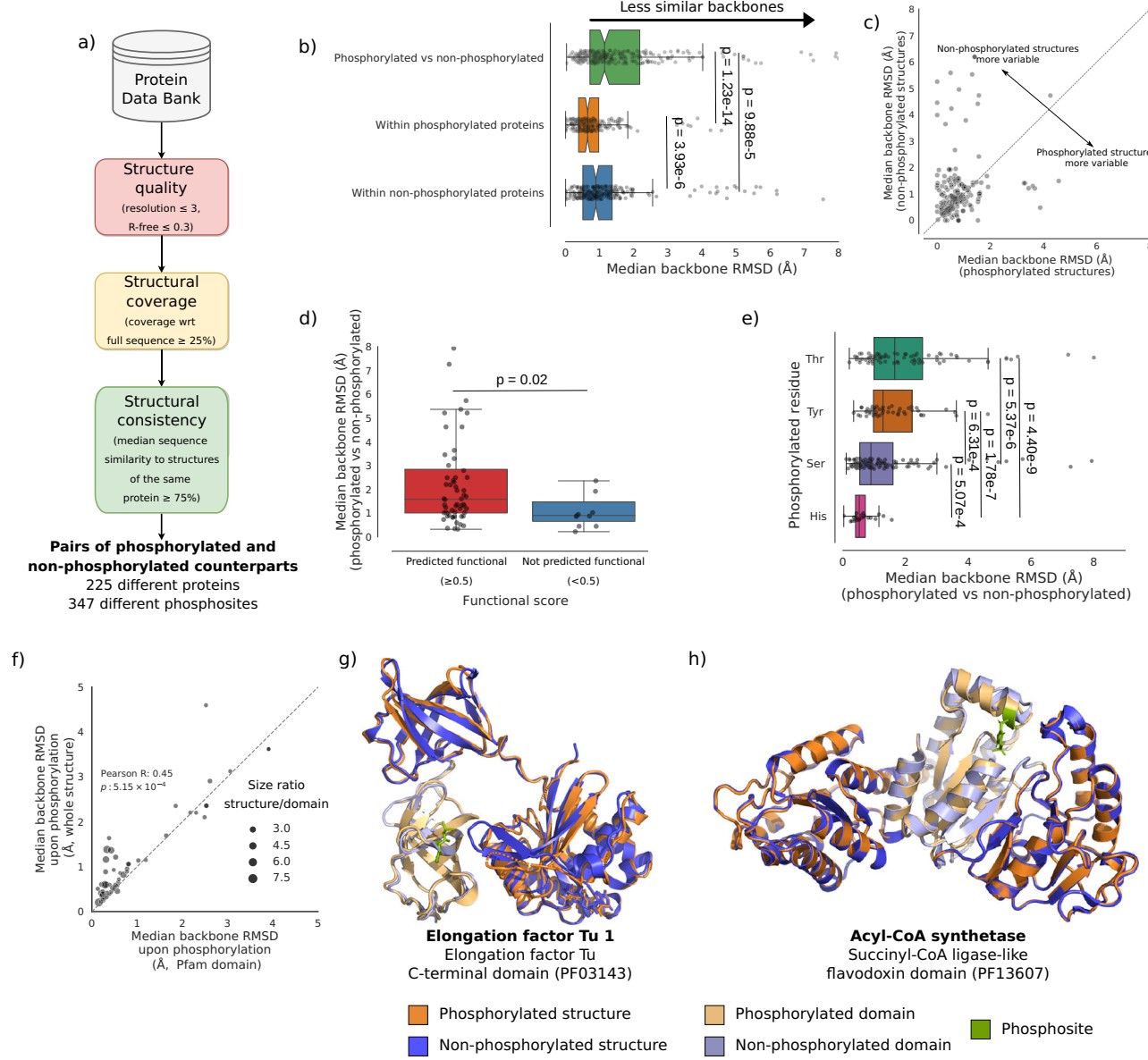

**Fig. 1 | Effects of phosphorylation on protein backbone conformation.**
**a** Retrieval and curation of phosphorylated and non-phosphorylated structures dataset. **b** Boxplot of conformational changes between phosphorylated and non-phosphorylated structures, and structural variability within each set (phosphorylated versus non-phosphorylated structures, $n = 290$ phosphosites; comparison within non-phosphorylated structures, $n = 260$; comparison within phosphorylated structures, $n = 210$; for each group, only cases where at least two structures were available were used). Points represent median changes per phosphosite. Notches indicate 95% confidence intervals; outliers beyond 8 Å not shown. **c** Scatterplot comparing median backbone RMSD per phosphosite within phosphorylated structures (x-axis) to non-phosphorylated counterparts (y-axis). The diagonal line represents the identity line; 63.95% of points are above the line. Only includes phosphosites with at least two structures in each state. **d** Boxplot of median backbone RMSDs between phosphorylated and non-phosphorylated structures according to predicted functionality (functional score ≥ 0.5, $n = 60$ phosphosites;

functional score < 0.5, $n = 11$). **e** Boxplot of median backbone RMSDs between phosphorylated and non-phosphorylated structures by phosphorylated residue (Ser, $n = 116$ phosphosites; Thr, $n = 85$; Tyr, $n = 57$; His, $n = 24$), sorted by median effect. **f** Comparison of median backbone RMSD per phosphosite within a Pfam domain (x-axis) to the whole structure (y-axis). Bubble size indicates the median size ratio between the whole structure and the Pfam domain. The *p*-value of the Pearson correlation is computed using a two-sided test. **g** Example showing greater conformational changes outside the phosphorylated domain (*E. coli* elongation factor Tu 1, UniProt ID: P0CE47). **h** Example showing conformational changes largely localized to the phosphorylated domain (*K. cryptofilum* acyl-CoA synthetase, UniProt ID: B1L3C9). In box-plots throughout the figure, the median is represented by the line inside the box. The box represents the interquartile range (IQR), while whiskers extend to the minimum and maximum values within 1.5 IQR. Points beyond the whiskers are outliers. Source data are provided as a Source Data file.

stratified the data, we found that phosphorylation within protein kinase domains is associated with significantly larger conformational changes (median RMSD: 1.51 Å) than in the rest of the dataset (median RMSD: 0.73 Å; one-tailed Mann–Whitney $U$ test, $U = 2842$, adjusted $p = 4.9 \times 10^{-11}$) (Supplementary Fig. 4b). We also compared the variance in backbone RMSD upon phosphorylation between the two groups and found no significant difference (two-tailed Brown–Forsythe test, $F = 0.002$, $p = 0.968$), indicating that the extent of conformational change associated with phosphorylation in kinase domains is not uniform. Notably, we observed that conformational changes associated with protein kinase domains are in general larger, regardless of phosphorylation status (Supplementary Fig. 4b). Overall, these results indicate that protein kinase domains are associated with above-average conformational changes in general, which somewhat inflates the observed conformational changes. This inflation is counter-balanced by the aforementioned lead author effect. Importantly, the core trends we report remain consistent regardless of the presence or absence of protein kinases in the dataset.

Interestingly, phosphosites predicted to be functional (functional score ≥ 0.5)[26] were associated with larger conformational changes than predicted non-functional sites (median backbone RMSD 1.58 Å in the former and 0.90 Å in the latter; Mann–Whitney $U$ test, $U = 456.5$, $p$-value = 0.02) (Fig. 1d). This suggests that the extent of conformational change may serve as a marker of phosphosite functionality.

We also examined whether phosphorylation of different residues is associated with varying degrees of conformational change. We found highly significant differences between residues (Kruskall–Wallis test, $H = 49.52$, $p$-value = $1.0 \times 10^{-10}$) (Fig. 1e). Threonine phosphorylation was associated with the largest conformational change (median RMSD = 1.65 ± 1.94 Å), while histidine showed the smallest effect (median RMSD = 0.52 ± 0.35 Å).

Finally, we asked whether structural changes were mostly localized or also affected distal regions (i.e., allosteric effects). To do so, we identified phosphosites within Pfam domains and compared the RMSD within the domain to the RMSD of the entire protein structure. To control for size-related effects, we only included examples where the overall structure was at least twice the size of the domain. We find that conformational changes tend to be greater in the overall structure than in the domain (Wilcoxon signed-rank test, $T = 309$, $p$-value = $6.4 \times 10^{-5}$) (Fig. 1f and Supplementary Fig. 6a), with significant changes between phosphorylated and non-phosphorylated overall structures (Wilcoxon signed-rank test, T = 185, $p$-value = $2.7 \times 10^{-2}$, Supplementary Fig. 6b), indicating that allosteric effects are common and significant. Additionally, while protein domains are often conceptualized as independent units[53,54], this result shows that phosphorylation within domains induces changes that propagate beyond and influence other regions of the protein. Fig. 1g, h show clear examples of phosphorylation inducing largely distal and largely local changes, respectively.

In summary, phosphorylation generally leads to small conformational adjustments that effectively appear to narrow down the conformational ensemble. Phosphorylated structures tend to be more uniform than non-phosphorylated ones, implying stabilization of a particular conformation. The extent of these conformational changes is indicative of the functional relevance of the phosphosite and varies depending on the specific residue phosphorylated. Furthermore, phosphorylation often exerts allosteric effects, well beyond the local environment of the phosphosite.

## Phosphorylated conformations are pre-existing and accessible regardless of phosphorylation

Motivated by our previous findings, we asked whether the structures of phosphorylated proteins occupy specific regions within the structural landscape shared by homologous proteins, and whether these are shared with non-phosphorylated structures. The availability of multiple experimentally solved structures for a protein, or homologous proteins, can be seen as a sampling of the different reachable conformational states[55]. We reasoned that, since protein domains are used as modular building blocks that can appear in different arrangements, they can provide a broader and more comprehensive sampling of structural space than simply studying whole homologous proteins. Thus, we focused on examining the conformational space of phosphorylated Pfam domains within our dataset. For each such domain, we retrieved all available structures from the PDB. After curating them, we created structural embeddings of the domains, from which we derived low-dimensional representations of the conformational space using principal component analysis (PCA) (Fig. 2a; see "Methods"). This procedure ensures that we can extract the dominant patterns in the data.

Upon clustering structures according to the low-dimensional representations, we found that phosphorylated structures strongly tend to be within clusters containing at least one non-phosphorylated structure. Specifically, in 15 out of 25 domains, 100% of phosphorylated structures were found within such clusters. For the remaining domains, either a large majority or a sizeable fraction of phosphorylated proteins are within one such cluster (Fig. 2b). A similar trend was observed when clustering was performed directly in the high-dimensional embedding space, confirming the robustness of this result (Supplementary Fig. 7). Manual inspection of the low-dimensional structural spaces clearly shows that phosphorylated proteins share specific clusters with non-phosphorylated ones (Fig. 2c and Supplementary Fig. 8). As an example, Fig. 2d shows representative structures in each cluster for the eukaryotic aspartyl protease domain (PF00026), illustrating the clear similarity of the phosphorylated structure to a specific subset of non-phosphorylated structures.

These observations provide evidence that, in general, the conformations adopted by phosphorylated proteins are pre-existing and inherently accessible, irrespective of phosphorylation. Note that in a small minority of domains, clusters of phosphorylated structures were found in regions without non-phosphorylated structures (Supplementary Fig. 8). These instances typically occur in domains with fewer solved structures or where there is a bias towards phosphorylated forms, suggesting that this is simply due to poorer sampling of the structural space, but it cannot be entirely ruled out that some such cases may represent genuine cases where the phosphorylated conformation is not accessible without phosphorylation. In summary, rather than a model where phosphorylation precedes structural rearrangements (i.e., induced fit[56]), our findings are more consistent with conformational selection[56–59].

## Phosphorylation induces local changes in residue fluctuations and alterations in modes of motion

As discussed above, phosphorylation often acts as an allosteric effector. It is now widely understood that protein dynamics are integral to the mechanisms of allosteric regulation[60,61]. This has led to the development of more quantitative, dynamics-based approaches to studying allostery, which may be particularly relevant to phosphorylation, as it typically induces small conformational changes. To explore changes in dynamics upon phosphorylation, we used ensemble normal mode analysis (eNMA) with atomistic elastic network models (ENMs). ENMs, widely used in allostery research[62–66], have been validated extensively for their consistency with both experimental and molecular dynamics (MD) results[64,67–69]. The ensemble NMA approach, which systematically models protein dynamics for all available protein structures for a specific comparison at once, allows us to directly compare ensembles of phosphorylated structures to ensembles of their non-phosphorylated counterparts. Our analysis focuses on three features predicted by eNMA: residue-wise fluctuations, normal modes of motion, and residue-residue couplings (Fig. 3a).

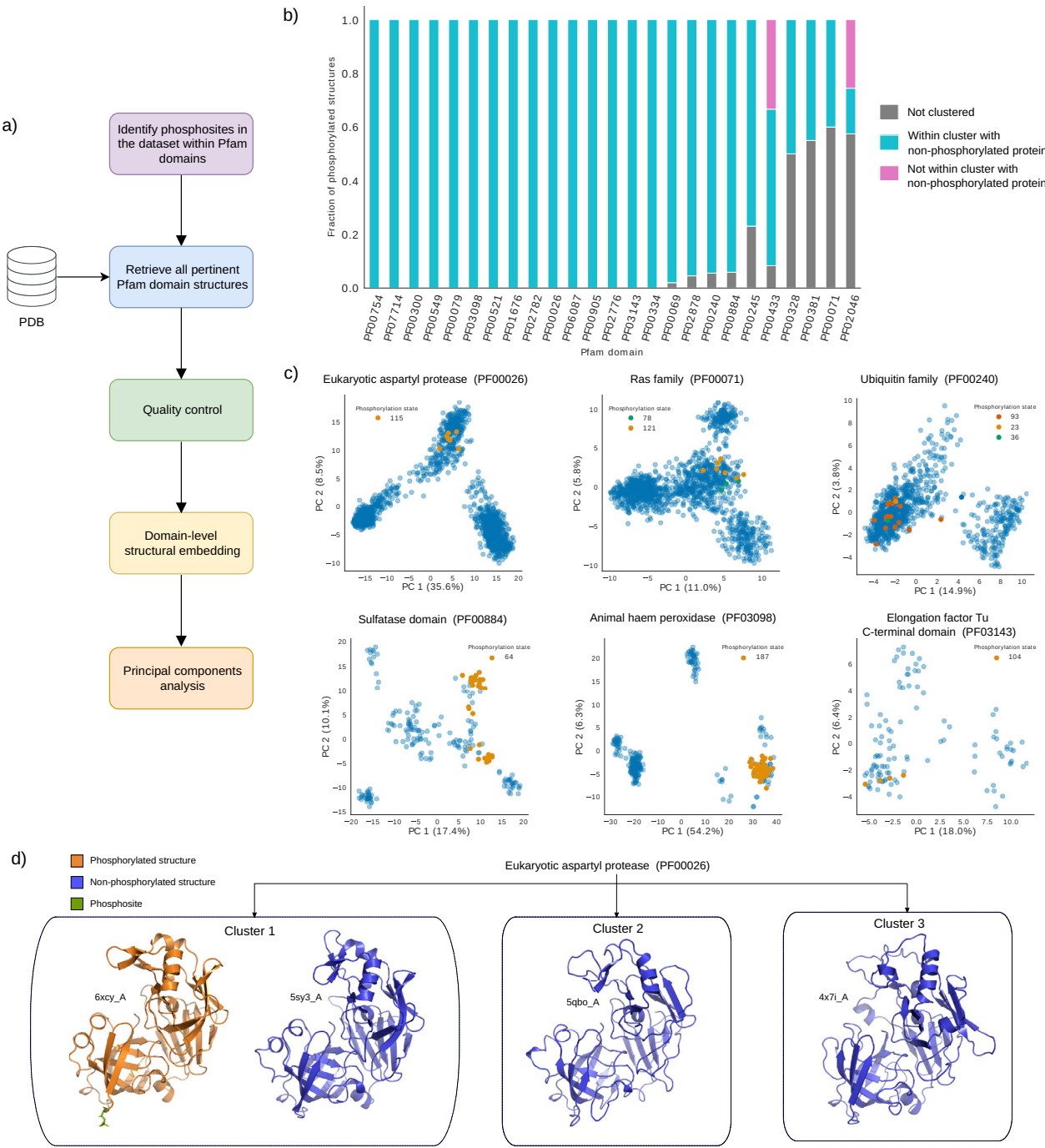

**Fig. 2 | Exploration of the conformational landscape of phosphorylated structures. a** Diagram depicting the procedure used to obtain low-dimensional representations of domain-specific structural spaces. **b** Stacked bar chart summarizing the proportion of phosphorylated structures, per domain, that are within a cluster containing at least one non-phosphorylated structure. Non-clustered proteins (gray) refers to structures labeled as not belonging to a specific cluster by HDBSCAN. **c** Scatterplots illustrating the clustering of phosphorylated and non-phosphorylated structures in six different low-dimensional, domain-specific representations. Blue dots represent non-phosphorylated structures, while other colors indicate specific phosphorylation states. Phosphosite indexes in the legend are assigned based on sequence alignments of the analyzed structures. **d** Examples of representatives structures from each cluster obtained for the eukaryotic aspartyl protease domain (PF00026). Source data are provided as a Source Data file.

We first analyzed global changes in predicted residue-wise fluctuations, which quantify the thermal movement amplitudes of individual residues. Our results showed an overall strong correlation between the fluctuation profiles of phosphorylated and non-phosphorylated proteins (Supplementary Fig. 9a). Significant changes in median flexibility occurred in only 10.9% of cases, with increased and decreased flexibility occurring in roughly equal proportions (Supplementary Fig. 9b). These findings indicate that phosphorylation seldom leads to global changes in protein flexibility and does not consistently affect it in one direction. Furthermore, no association was found between changes in overall fluctuations and fluctuation profile correlations (Supplementary Fig. 9c), implying that phosphorylation may primarily affect local dynamics instead (illustrated in Fig. 3b).

To distinguish significant local changes associated to phosphorylation from mere noise caused by structural variation, we first conducted a comparative analysis of per-residue fluctuations between phosphorylated and non-phosphorylated states to systematically assess the impact of phosphorylation on local dynamics. To set a reliable cutoff for significance, we first evaluated the intrinsic variability in the dataset by computing the absolute value of per-residue fluctuation differences for all pairwise comparisons between phosphorylated and non-phosphorylated proteins, and from there derived a stringent cutoff value (see "Local comparison of residue-wise fluctuations" subsection under the "Methods"). We found that 82.8% of phosphorylation events were associated with at least one significant local peak in fluctuation differences between phosphorylated and non-phosphorylated structures, with a median of three significant peaks per event and a median peak height of 0.77 Å$^2$ (Supplementary Fig. 9e, f). To further assess changes, we compared per-residue fluctuations using the Wilcoxon signed-rank test, discovering significant changes in 80.5% of cases. Combining these criteria (at least one significant peak and an adjusted $p$-value $\leq 0.05$), we identified significant changes in 69.5% of cases. These findings suggest that phosphorylation frequently alters local thermal fluctuations; however, the direction of these changes is not consistent, with local fluctuations increasing and decreasing in roughly equal proportions across phosphorylation events (Fig. 3c and Supplementary Fig. 9d). Finally, we assessed the extent of changes in residue fluctuations in the neighborhood of the phosphosite, finding that these are clearly insufficient to drive the observed trends (Supplementary Fig. 10), in agreement with Subhadarshini's et al. conclusions[70].

Changes in local fluctuations result from modifications in the underlying modes of motion, which are coordinated patterns where all residues oscillate at the same frequency. These modifications can indicate functional changes. We assessed the conservation of normal modes between phosphorylated and non-phosphorylated proteins across four frequency regimes: global modes ($k = 1$–3), low frequency (LF) modes ($k = 4$–20), low to intermediate frequency (LTIF) ($k = 21$–60), and high frequency modes ($k > 60$). Previous studies have shown that lower frequency modes are more evolutionarily conserved, involve larger groups of residues[71,72] and are often linked to allostery and functional motions, indicating greater biological relevance[63,64,73–77]. These modes can be modified by factors such as protein-protein interaction or ligand binding[78,79].

Our analysis revealed broad variability in changes to the 20 slowest modes across phosphorylation events, prompting us to to stratify them into three subsets based on the extent of these changes (Fig. 3d). Notably, even the subset with the smallest changes (comprising 78% of phosphorylation events) exhibited significant changes in normal modes (Supplementary Fig. 11), suggesting phosphorylation generally fine-tunes protein motions, typically distinguished by LF motions. A clear example of changes within the subset with smallest changes is elongation factor Tu 1 (EF-Tu 1), which undergoes inactivation via restricted dynamics upon phosphorylation[35]. Our analysis shows clear static areas in the phosphorylated form (LF mode 5, Fig. 3e). More substantial changes are seen in other subsets, particularly in global and LF modes. For example, aurora kinase A, which becomes activated upon phosphorylation[80], shows a general twisting motion in its phosphorylated form (global mode 2, Fig. 3f). These observations indicate dynamic alterations linked to phosphorylation, with potential implications for protein function.

Overall, our results indicate that the role of phosphorylation in modulating residue fluctuations is primarily through significant changes in local thermal fluctuations, rather than global changes. The absence of a consistent trend towards increased or decreased flexibility suggests highly context-dependent effects. These changes in fluctuations are driven by changes in normal modes, supporting a model of fine-tuned dynamic regulation in most cases.

## Phosphorylation modifies residue-residue couplings in diverse but infrequent ways

Couplings between residue motions underlie the cooperativity that is key to biomolecular allostery[60]. Correlation analysis allows us to detect the presence of couplings between spatially distant sites and the molecular elements responsible for transmitting signals between them[60]. One way in which phosphorylation may have an effect on protein function is by altering couplings between residue motions. For example, it has been shown that, together, activation loop phosphorylation and substrate binding trigger uniform dynamics across the MAP kinase p38. This synchronization allows ATP recruitment[33]. This possibility, however, is seldom considered.

Here, we focused on studying global effects of phosphorylation on predicted residue-residue couplings. In order to systematically investigate this effect, we computed residue-residue mutual information matrices for all cases where we could perform ensemble NMA. The similarity between mutual information matrices was assessed using the $RV_2$ coefficient. We find that predicted residue couplings tend to be highly similar between phosphorylated and non-phosphorylated structures, with 87% of phosphorylated versus non-phosphorylated comparisons having a median $RV_2$ coefficient $\geq 0.90$ (Fig. 4a). This finding indicates that, while phosphorylation can significantly alter residue motion couplings, such a mechanism appears to be uncommon.

Another way in which phosphorylation might alter the residue couplings is simply by an overall increase or decrease in the couplings, leading to tense or relaxed states, respectively[60]. We compared the overall difference in mutual information between states to the corresponding $RV_2$ coefficients, and found that phosphorylation can indeed alter dynamic couplings on both axes (Fig. 4b). In fact, such changes often go together. From this comparison, we distinguished three possible types of changes upon phosphorylation: relaxation (overall decrease in mutual information), tensioning (overall increase), and rewiring (small overall changes in the overall mutual information concomitant with lower correlations between the phosphorylated and non-phosphorylated states). Fig. 4d showcases three interesting examples of alterations in residue couplings, with phosphosites shown in stick representation, one for each category. An especially illustrative case is the rewiring example, where residue-residue couplings are strikingly reshaped within a disordered region containing the phosphosites. This region includes many charged residues, both positive and negative, and phosphorylation introduces additional negative charge (Supplementary Fig. 13). These changes in the local electrostatic environment likely drive the substantial reorganization of dynamic couplings observed.

In summary, our findings indicate that phosphorylation can significantly impact global dynamic residue couplings across two different axes, despite this being a relatively infrequent phenomenon. This mechanism appears to have been largely overlooked in previous research.

## Phosphosites display low levels of mechanical coupling with protein functional sites

Allostery involves transmitting a perturbation from a regulatory site to a distal functional site, thereby modulating protein activity. This process can trigger a cascade of localized deformations between specific regions of the protein, as described by the domino model of allosteric signal transduction[60,81]. Identifying such deformations can be especially challenging for phosphorylated proteins, because most proteins do not display large global conformational effects upon phosphorylation, as shown above. To capture local deformations, we employed an elasticity-based approach, Protein Strain Analysis (PSA), focusing on identifying the mechanical couplings between phosphosites and functional sites that may mediate allosteric signal transduction[82,83]. Unlike RMSD, strain analysis is alignment-independent and highlights elastically deformed regions (rather than displacement and rotations)

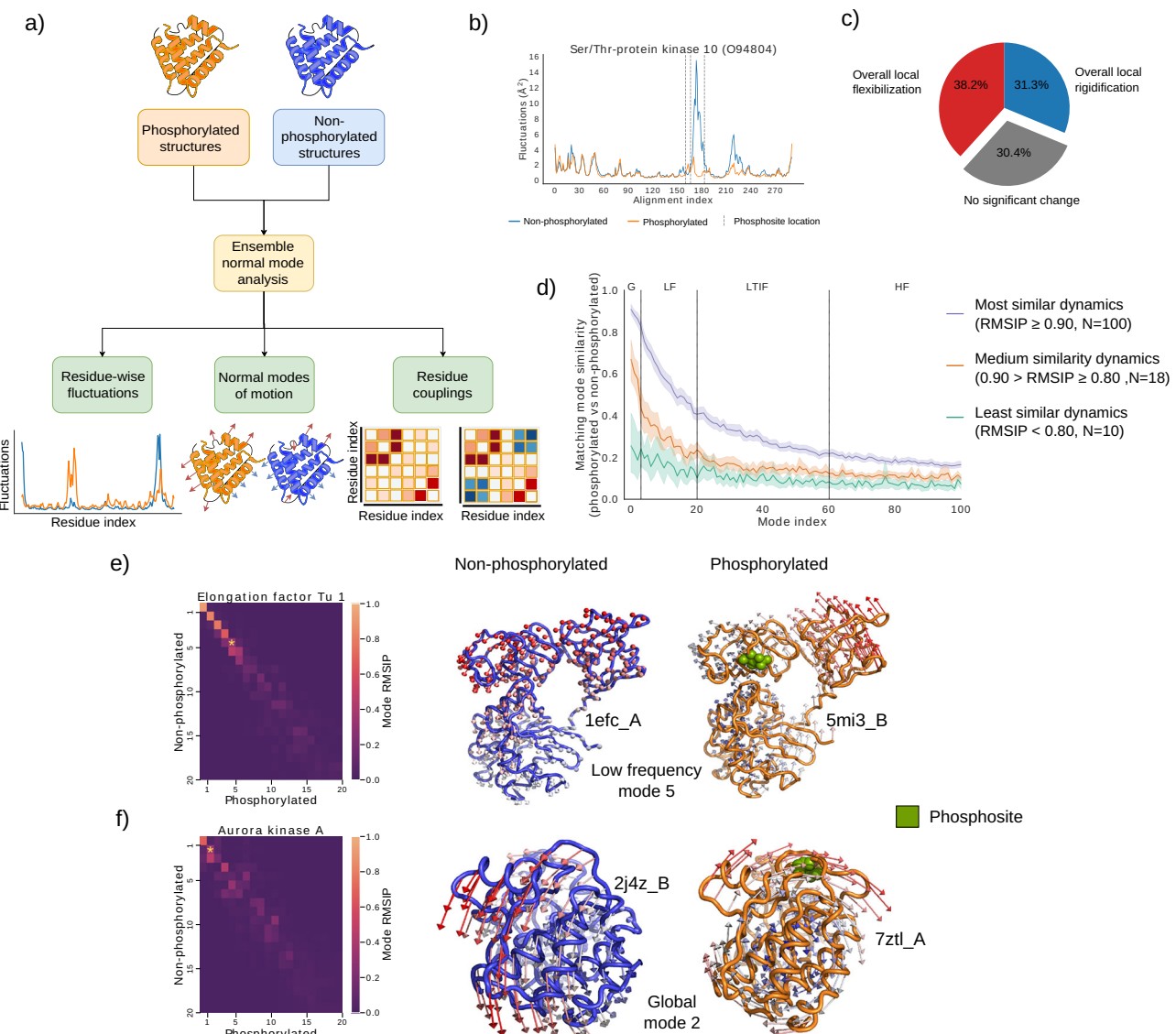

**Fig. 3 | Effects of phosphorylation on predicted residue-wise fluctuations and normal modes of motion. a** Diagram illustrating various aspects of protein dynamics under study. **b** Example of clear local changes in residue fluctuations upon phosphorylation. **c** Pie chart categorizing the types of local changes in dynamics. **d** Similarity of matching normal modes between phosphorylated and non-phosphorylated proteins over four different frequency windows (from slowest to fastest), stratified by overall similarity on the first 20 modes. G global, LF low frequency, LTIF low to intermediate frequency, HF high frequency. Shaded bands indicate 95% confidence intervals of the median similarity. **e**, **f** Two different examples of changes in normal modes. Above: elongation factor Tu 1 from *E. coli* (UniProt ID P0CE47). Below: human aurora kinase A, (UniProt ID O14965).

Heatmaps show all pairwise similarities (as RMSIPs) between the 20 lowest frequency modes; yellow asterisks highlight a normal mode of the same relative frequency but with clear differences between phosphorylated and non-phosphorylated structures. Vector field plots show the highlighted normal modes of the same relative frequency for non-phosphorylated (left) and phosphorylated structures (right). Arrow length proportional to the magnitude of fluctuations (for clarity, scaled differently above and below). Phosphosite indicated in green spheres. The protein icon (credited to Database Center for Life Science) and the heatmap icon (credited to Chenxin Li) in panel a are from Bioicons (bioicons.com) and are under the CC BY 4.0 license. Source data are provided as a Source Data file.

between a reference structure and a deformed structure. Previous studies have shown elevated strain in functional and allosteric sites, as well as in the regions connecting them, supporting a mechanism through which perturbations can propagate between allosteric and functional sites[83–85]. Here, we applied this method to compare phosphorylated and non-phosphorylated structures, focusing on identifying proteins for which both functional sites and phosphosite display a signature of structural deformation (Fig. 5a).

We first evaluated overall trends in the dataset. To test whether there is mechanical coupling between phosphosite and functional sites, we estimated the local mean strain in the neighbourhood of the phosphosite (in a window of ± 2 residues), in the functional site and the

background (Fig. 5b). Our analysis revealed an overall significant increase in strain in phosphosite vicinities compared to the background (Mann–Whitney $U$ test, $p = 9 \times 10^{-7}$, $U = 20821$, C$LES = 0.65$), whereas the strain in functional sites was slightly lower than in the background ($p = 2 \times 10^{-7}$, $U = 32073$, C$LES = 0.37$). Also, comparisons between data and bootstrapping showed that the strain at the phosphosite was significantly larger than expected by chance. This suggests that phosphorylation tends to induce significant local deformation near the phosphosite, while functional sites generally remain relatively stable.

Subsequently, we examined mechanical coupling in individual proteins, comparing the mean strain in the data described above to the bootstrap samples. Our results show that 27% of phosphosites display

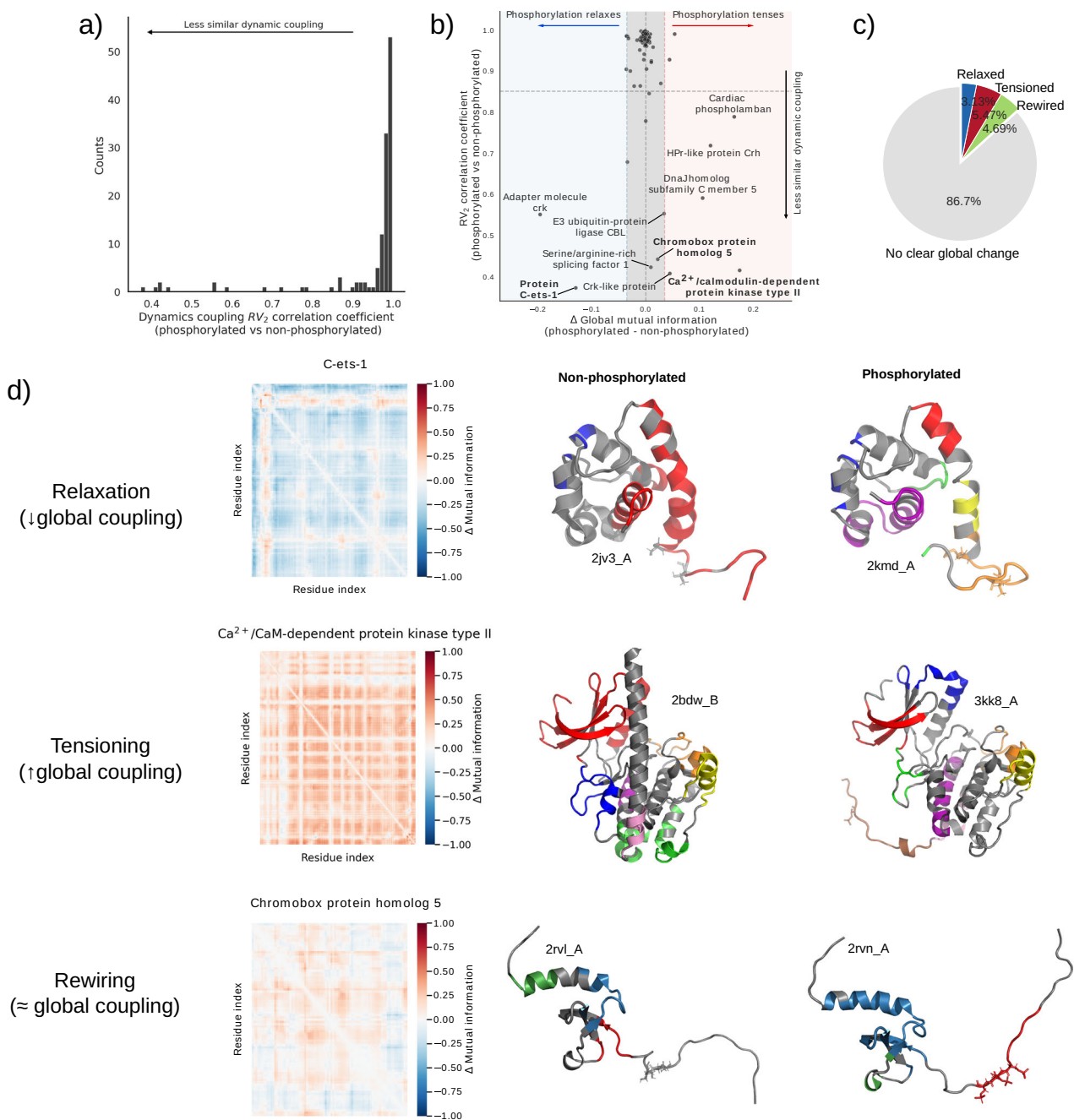

**Fig. 4 | Effects of phosphorylation on predicted residue-residue couplings.**
**a** Histogram of median $RV_2$ coefficients comparing phosphorylated and non-phosphorylated mutual information matrices, indicating the degree of global change in residue-residue couplings upon phosphorylation. **b** Scatterplot comparing the degree of global change in residue-residue couplings, as measured by the $RV_2$ coefficient (y-axis) to the direction of overall change in mutual information for the entire structure (x-axis) upon phosphorylation. Dashed lines on the x-axis represent the median value ± one standard deviation. **c** Pie chart summarizing the observed changes in residue-residue couplings upon phosphorylation. **d** Three examples of clear and significant global changes in residue-residue couplings upon

phosphorylation (highlighted in bold in (**b**)). Heatmaps on the left show the median difference in residue couplings between phosphorylated and non-phosphorylated structures. Residues shown in sticks indicate phosphorylated residues. Note that the phosphosite in 2bdw_B is in the pink cluster and buried between two helices, thus partially occluded in this view. Residues with the same colour indicate clusters of highly coupled residues; gray indicates residues not part of any cluster. For clarity, in the case of $Ca^{2+}$/CaM-dependent protein kinase II, only clusters with more than seven residues are indicated, due to the higher number of clusters. Source data are provided as a Source Data file.

high deformation compared to the bootstrap samples, whereas only 12% of functional sites do (Fig. 5c). Furthermore, only 5% of phosphosite-functional site pairs were found to be co-strained (Supplementary Data 3). The phosphosite-functional site pairs range from close distances to 35 Å apart, hinting at both direct and allosteric control depending on each case (Fig. 5d). This limited mechanical

coupling between phosphosites and functional sites is further supported by the low correlation between strain values in these two regions (Supplementary Fig. 14), indicating largely independent deformation patterns.

We then investigated the functional relevance of proteins where phosphosite and functional sites are co-strained (Fig. 5c).

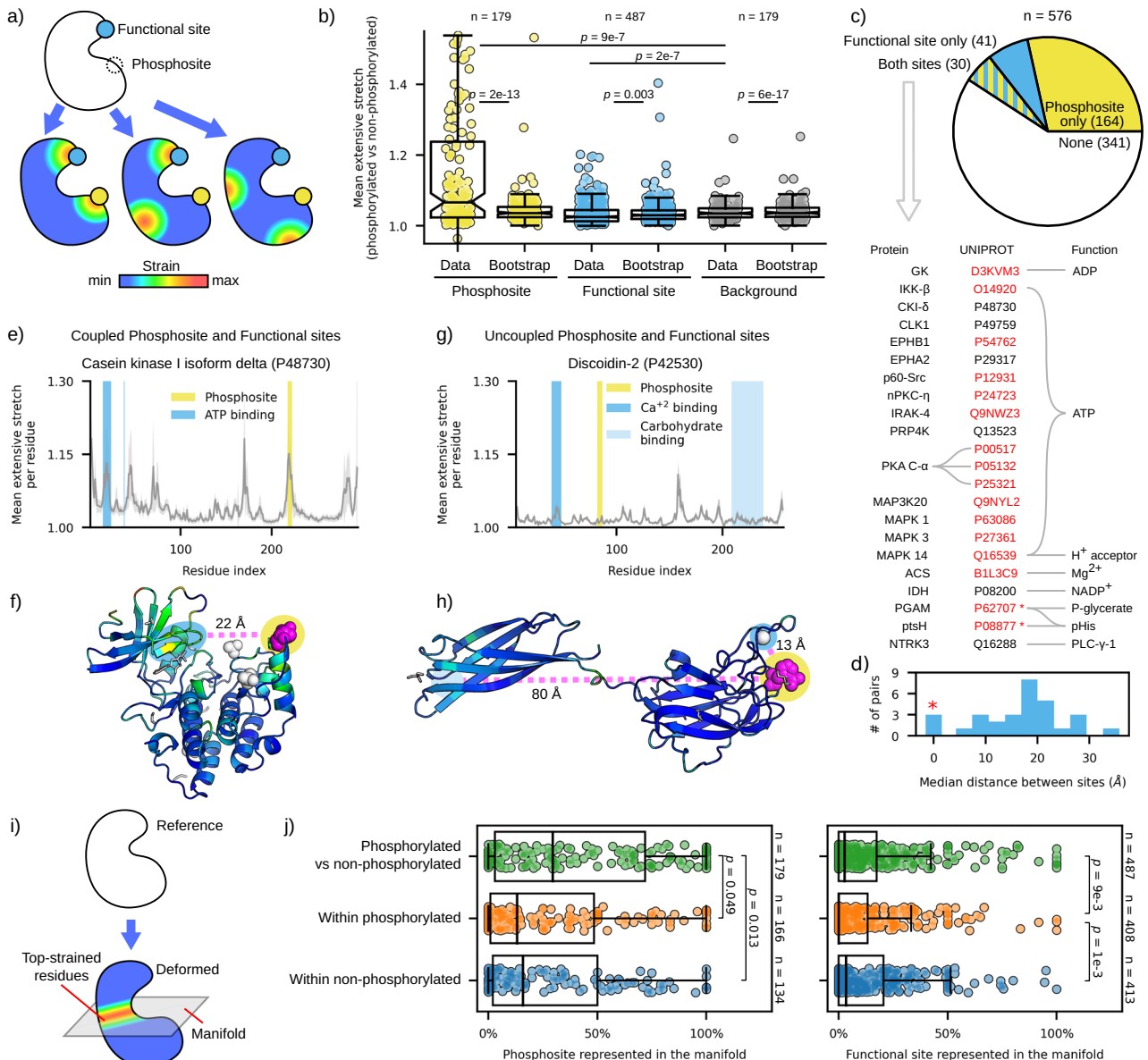

**Fig. 5 | Protein strain analysis reveals local deformations associated with phosphorylation. a** Schematic showing mechanical strain from pairwise comparisons of phosphorylated vs. non-phosphorylated structures, focusing on overlap with phosphosites, functional sites, and background (i.e., any other residues). **b** Boxplot comparing strain across different protein regions for real data vs. bootstrap samples (randomized strain distributions). **c** Overlap between highly strained phosphosites and highly strained functional sites. Proteins with clear co-straining in both sites are listed, with red-highlighted UniProt IDs indicating literature-supported functional roles for phosphorylation. Asterisks mark phosphosites within functional sites. **d** Histogram of distance between co-strained phosphosite and functional site. **e, f** Predicted mechanical coupling between phosphosite T220 and ATP binding site in human casein kinase I, isoform delta, with extensive stretch per residue averaged over multiple pairwise comparisons, with the standard deviation represented in the shaded bands. The comparison shown in the structure is 7P7F_B vs 4HNF_A. **g, h** Example of no mechanical coupling

in discoidin-2 from *Dictyostelium discoideum*, which binds calcium ions and carbohydrates in distinct domains, showing no strain increase either at the phosphosite (H84) or at ligand-binding sites. The comparison shown in the structure is 2VME_B vs 2VMC_A. Strain from phosphorylated vs. non-phosphorylated structures is color-coded by stretch magnitude (1.0–1.2); phosphosites in magenta, ligands in white. Shaded bands indicate standard deviation. **i** Schematic representation of strain mapping in a structural comparison, highlighting the high-strain manifold. The reference structure (in white) is contrasted with the deformed structure (coloured according to the strain magnitude). **j** Representation of phosphosite vicinity and functional site in the high-strain manifold across groups. Statistical significance was determined using two-tailed Mann–Whitney *U* tests. In box-plots throughout the figure, the median is represented by the line inside the box. The box represents the interquartile range (IQR), while whiskers extend to the minimum and maximum values within 1.5 IQR. Points beyond the whiskers are outliers. Source data are provided as a Source Data file.

Interestingly, many of these proteins display a known relationship between the particular phosphosite found to be strained and the protein's molecular function. In many cases, the phosphosites are directly involved in the phosphotransferase, phosphatase, isomerase, and ligase activities, which explains the overrepresentation of ATP binding activity (hypergeometric test, $p = 7 \times 10^{-12}$). Fig. 5e–h

showcases two proteins with contrasting results in terms of their functional mechanical properties. For casein kinase I-delta, we find that both the ATP binding pocket and the phosphosite are co-strained and co-localized in the high-strain manifold, despite being far apart in the phosphorylated form. In this example, we observe that residues in $\beta$-sheets and their connecting loops in the vicinity of

the ATP binding site consistently display peaks of high strain, Fig. 5e. This co-deformation pattern hints at allosteric regulation between sites. In contrast, discoidin-2, a galactose-binding lectin, shows a smaller extent of deformation throughout the protein structures (Fig. 5g), with the largest strain localized at the linker loop between the two β-sheet domains (Fig. 5h). Therefore, our analysis finds no evidence of mechanical coupling between phosphosite and functional site in this particular case.

A complementary approach useful for understanding deformation patterns relies on the characterisation of the high-strain manifold. In essence, this manifold consists of the set of residues within the protein structure that concentrates most of the protein's deformation between two conformations (Fig. 5i and Supplementary Fig. 15a, b). Previous studies have shown both binding sites and allosteric sites co-localized to a quasi-2D manifold of high deformation, suggestive of signal propagation between them, in glucokinases and aspartate carbamoyltransferase[84,85]. Given this, we implemented this analysis to investigate the general and local properties of the high-strain manifold in phosphorylated structures. Interestingly, the comparison among phosphorylated, non-phosphorylated, and between these groups shows no detectable difference in relative size and dimensionality of this manifold (Supplementary Fig. 15c), matching the values reported in ref. 84. This suggests that, at a whole protein scale, deformation patterns remain unaltered upon phosphorylation. We then tested whether the specific residues belonging to the high-deformation manifold are the same across comparisons. More specifically, we computed the frequency in which the phosphosites and functional sites were represented within the manifold in each group comparison. We found that the representation of phosphosite in the high-strain manifold between groups is higher than within groups. The functional sites, however, were scarcely represented regardless of the group comparison (Fig. 5j). We also tested which phosphosite-functional site pairs were detected as significantly represented in the manifold, resulting in 7.5% of the total number of pairs (Supplementary Fig. 16). Overall, these results agree with the mean strain comparison, showing that phosphosites tend to be more strained upon phosphorylation, while mechanical coupling to functional sites remains relatively limited.

In summary, our findings reveal that phosphorylation clearly induces significant local strain near phosphosites, whereas functional sites remain more stable. We find a small, but significant, subset of cases where the phosphosite and a functional site are co-strained, indicative of mechanical coupling between them. In such cases, we found extensive support in the literature suggestive of allosteric control between the phosphosite and the coupled functional site, particularly in ATP binding pockets. Our analysis of high-strain manifolds further supports this, showing that phosphosites are more often represented within the manifold than functional sites. These results suggest that allosteric regulation through mechanical coupling between phosphosites and functional sites is the exception, rather than the rule.

## Discussion

Here, we conducted a species-independent, comparative structural analysis of the mechanisms by which phosphorylation regulates proteins, inspecting changes in conformation, dynamics, and mechanical strain between non-phosphorylated structures and their phosphorylated counterparts. By leveraging redundancy in the PDB, our analysis accounts for intrinsic structural variability. As the function of vast numbers of phosphosites remains uncharacterized, we focus on exploring the application of computationally inexpensive methods that could potentially be used to screen the effects of many phosphosites. While available sample sizes in the PDB for many of the hundreds of existing PTMs are likely insufficient, this methodology is nonetheless broadly applicable to studying the effects of other PTMs, such as acetylation.

We identified different general trends of regulation by phosphorylation. Regarding protein structure, we find that conformational changes upon phosphorylation are generally small and stabilizing, consistent with previous research[42]. Additionally, allosteric effects are commonly observed. The extent of conformational changes depends on the phosphorylated residue, with threonine phosphorylation associated to the largest changes on average, and on the predicted functionality of the phosphosite. This association between extent of conformational change and phosphosite functionality suggests that structure prediction tools that can incorporate phosphorylation[86] may serve as part of a screening to identify functional phosphosites. We emphasize, however, that a small conformational change does not imply lack of phosphosite functionality, or vice versa. Both orthosteric and allosteric effects may take place without observable or minimal backbone changes[87,88].

Our findings suggest that phosphorylation primarily selects an existing conformation from an underlying conformational ensemble, rather than inducing an otherwise inaccessible state. In this framework, all possible conformations of a protein are populated according to their energies; lower energy conformations are more populated than higher energy ones. Allosteric effectors, such as ligands or, in our case, phosphorylation, cause shifts in the relative populations of the existing conformations in a conformational ensemble. This conformational selection, in turn, is what triggers changes in, e.g., catalytic activity in the overall protein population[89,90]. Note that factors such as a large sample size of non-phosphorylated species or the overall structural similarity between structures of a specific domain alone cannot account for our observations, as the outcomes are determined by the underlying mechanism of action. If phosphorylation consistently triggered conformational changes in accordance with the induced fit model (where the phosphorylated conformation would be inaccessible in the absence of phosphorylation), the phosphorylated structures would be distinctly isolated in the low-dimensional representation of Pfam domains (Fig. 2c), irrespective of these factors. PTMs are sometimes described in the literature as increasing the diversity of the proteome[91]. While our study focuses on phosphorylation, the conformational selection framework is broadly applicable to various molecular interactions, such as protein-ligand or PPI. Based on our results and the general applicability of conformational selection, we predict that, in some cases, PTMs may more accurately be viewed as altering the distribution of an already existing diversity, rather than increasing proteome diversity.

We identified pervasive changes in predicted local residue fluctuations, suggesting an important role for entropic allostery in regulation of protein structure by phosphorylation. This is, in general, now thought to be a highly prevalent allosteric mechanism[29]. Such changes in fluctuation amplitudes can mediate allosteric shifts[92,93] and imply alterations to the underlying modes of motion, which can mediate functional changes. Interestingly, we also found that in ~13% of cases, phosphorylation causes global changes to predicted residue-residue couplings, an effect that has been largely underappreciated in previous literature. We suggest that alterations in residue-residue couplings upon phosphorylation should not be overlooked, as they may indicate changes in the transmission of allosteric signals[60]. Our findings contradict those of Subhadarshini et al.[70], whose analysis found no effect of phosphorylation on predicted residue couplings. This discrepancy may be attributed to the small sample size in their study (17 pairs of phosphorylated and non-phosphorylated structures) and the use of the unmodified RV coefficient, which inflates matrix correlation values[94] and therefore underestimates the degree of change.

Allosteric modes of action are largely uncharacterized and have eluded a comprehensive, predictive biophysical description[29,95]. We found evidence of mechanical coupling between phosphosites and functional sites in a relatively small, but nonetheless significant, subset of cases. Several factors may account for the limited size of this subset.

First, the consequences of phosphorylation may be mostly kinetic instead of structural, e.g., phosphorylation mainly narrows the protein's conformational landscape, consistent with the conformational selection framework[57]. This is supported by our observation that phosphorylated structures tend to exhibit greater uniformity and cluster in specific areas of the conformational landscape. Another example of such kinetic effects is the alteration in the protein's vibrational profile at different sites, as suggested by the violin model[60,81]. This model is thought to better describe distal allosteric regulation; however, inferring the impact of changes in vibrational patterns from protein structures remains challenging. Second, the high conformational variability between structures of the same protein poses a challenge to analyse local conformational changes for large datasets. A more targeted approach using manually curated structural datasets may be better suited to investigate couplings for a specific protein. Finally, individual phosphorylated residues may have little effect on inducing conformational changes. Therefore, a strategy that integrates data from multiple phosphosites at once could enhance the sensitivity of future analyses.

While we observe some general trends (e.g., phosphorylated proteins tend to be more uniform), it is clear that, as previously noted[41], the effects of phosphorylation are rather diverse (e.g., phosphorylation increases structural diversity in some cases). We hypothesize that these outcomes are highly dependent on the local context: for example, previous research shows that serine phosphorylation at the N-terminus of an $\alpha$-helix stabilizes the helix via favourable electrostatic interactions with the helix dipole, whereas phosphorylation at the C-terminal end destabilizes the helical structure[96]. We expect further insights will be gained by studying the local properties of the phosphosite, such as energetic frustration[30,97] or its chemical environment[98], and correlating these factors with the observed changes upon phosphorylation.

Our study has several limitations, some of which arise from biases in the PDB. The PDB largely consists of X-ray crystallography structures, which, due to entropic effects, are harder to obtain as the conformational flexibility of a structure increases. For instance, if phosphorylation causes drastic transitions between ordered and disordered states, crystallizing one of these states may become exceedingly difficult or even impossible. Consequently, the absence of such structures in the PDB would render our findings more conservative. Similarly, this also means our study cannot describe changes that take place in intrinsically disordered proteins upon phosphorylation, which requires alternative approaches[99,100]. Another bias is the lead author effect, where structures generated by the same research group tend to be more similar, further contributing to our results probably being an underestimate. More generally, we suggest that efforts mining the PDB for comparative structural analyses should account for this effect. Finally, the PDB is biased towards functional, or likely to be functional, phosphosites; it is natural that research efforts will be directed towards them. A better understanding of phosphosite functionality will require solving more structures phosphorylated at positions that do not appear to be functional, or further development of structure prediction methods that can incorporate PTMs[86]. Further studies may also take into account more complex factors, such as possible allosteric effects involving the solvent[93,101,102].

Many cellular functions are carried out by protein circuits, where proteins modify each other's activity, localization, or stability. This can be achieved by PTMs, as in certain circadian clocks driven exclusively by phosphorylation[103]. Yet, despite this interest in developing circuits that can be controlled by PTMs, which has nonetheless materialized in a few recent examples[104,105], most efforts in synthetic biology have focused on the design of circuits controlled by transcriptional regulation, due to the relative ease of engineering circuits controlled by transcription factors. To achieve the aim of PTM-driven circuits, we will need to develop general strategies to rationally engineer novel phosphosites and, more generally, PTM sites, not seen in nature[106]. We expect that further research on how phosphorylation influences protein structure and function will contribute to the development of such strategies, unlocking a largely unexplored layer of regulation in synthetic biology.

## Methods

In box-plots throughout the manuscript, the median is represented by the line inside the box. The box represents the interquartile range (IQR), while whiskers extend to the minimum and maximum values within 1.5 IQR. Points beyond the whiskers are outliers.

### Structural data collection and curation

We retrieved phosphorylated structures and their non-phosphorylated counterparts from the PDB by querying the PDB relational database[107,108] for pairs of structures of the same protein (i.e., the structures must share the same UniProt ID) where the same residue is phosphorylated in one structure but not the other (i.e., one residue has the code SER, THR, TYR, or HIS in one and SEP, TPO, PTR, HIP, or NEP in the other, respectively). The raw dataset contained 484 different proteins and 812 different phosphosites.

Structures resolved by X-ray crystallography or electron microscopy were kept only if they had a resolution ≤ 3 Å and R-free ≤ 0.3. Furthermore, we also ensured that the structures for each protein cover a reasonably sizable fraction of the whole protein and are consistent with each other. To do so, we created a multiple sequence alignment for each protein using Clustal Omega v1.2.4 with default parameters[109]. As input, we provided the full sequence from UniProt and the sequences derived from each structure. We then calculated the coverage of each structure with respect to the full length protein sequence and the median sequence similarity of each aligned structure-derived sequence to the rest. The purpose of the latter metric is to serve as a measure of how consistent a given structure is to the rest, in terms of the part of the sequence they cover. We kept only structures with a coverage with respect to the full sequence ≥ 0.25 and a median similarity ≥ 0.75. While the coverage threshold allows us to keep relatively small parts of the whole protein, we observed that, in combination with the median similarity cutoff, this was sufficient to study domains or the structured parts of otherwise highly disordered proteins. Fig. 1a shows a summary of the curation process; Supplementary Fig. 1 shows intermediate steps and a summary of the filtered dataset.

The filtered dataset contained 225 different proteins and 347 different phosphosites. For each phosphosite, there was at least one corresponding phosphorylated and non-phosphorylated structure. Of these 347 phosphosites, 274 (~79%) are single-site phosphorylations. The rest are multisite phosphorylations (i.e, proteins are phosphorylated at multiple sites), which are either grouped or excluded depending on the analysis. For each phosphosite, there was at least one corresponding phosphorylated and non-phosphorylated structure (Supplementary Fig. 1c). In cases where multiple structures are available, this is leveraged to take into account intrinsic structural variation.

### Retrieval and curation of Pfam domain-level structural data

We scanned the protein sequences in the filtered dataset for the presence of Pfam domains using InterProScan v5.65–97.0[110]. We then filtered out Pfam domain hits with an $E$-value > 0.01 and selected hits that encompass at least one phosphosite in our dataset. Amongst these, we identified 65 different Pfam domains and 239 phosphosites within a domain (68.9% of all phosphosites in the dataset).

We then retrieved all available structures in the PDB that contain at least one hit for each Pfam domain, using information about Pfam domains available in the SIFTS resource[111]. We extracted the part of the structure containing the domain using the domain boundaries defined by SIFTS using pdb-tools v2.5.0[112]. Structures with a resolution above

≥ 3 Å, with a domain coverage ≤ 0.75 or whose domain boundaries are ambiguously defined in SIFTS were discarded. Finally, we filtered out structures that contain > 10% of residues where only the coordinates for the Cα are present. After these filtering steps, we have 55 different Pfam domains for which we have at least 10 different structures.

## Backbone comparison

In order to compare the backbones of both whole protein structures and domain structures, we computed α-carbon RMSDs between protein structures by superimposing them using singular value decomposition as implemented in Biopython v1.76[113]. Matching α-carbons were defined by global pairwise alignment (as implemented in BioPython v1.76) of the corresponding sequences using the BLOSUM62 matrix[114] and a gap open penalty and gap extend penalty of −10 and −0.5, respectively. In the event that an NMR structure is involved in the comparison, and in order to account for its conformational variation, we computed the RMSD to all different models and take the median.

For each phosphosite, we perform three sets of pairwise comparisons in the same fashion as[42]: all pairwise comparisons between phosphorylated structures, all pairwise comparisons between non-phosphorylated structures, and all pairwise comparisons between phosphorylated and non-phosphorylated structures. Note that these comparisons are exclusively conducted between different experimentally solved structures of the same protein (i.e., they must share the same UniProt ID), and never involve distinct but homologous proteins. In this way, we quantify both the conformational changes taking place upon phosphorylation and the intrinsic variability found among phosphorylated and non-phosphorylated proteins. Differences between the distribution of median RMSDs between phosphorylated and non-phosphorylated structures ($n = 290$) and within group comparison distributions ($n = 260$ for the comparison within non-phosphorylated structures, $n = 210$ for the comparison within phosphorylated structures; we use only cases where at least two structures were available per comparison) were assessed with one-tailed Mann–Whitney $U$ tests. P-values were adjusted using the Benjamini–Hochberg procedure[115] (FDR < 0.05) using statsmodels v0.13.5[116]. A two-tailed Wilcoxon signed-rank test was used to test differences between phosphorylated structures and their non-phosphorylated counterparts ($n = 197$, including only cases where at least two phosphorylated and non-phosphorylated structures were available per phosphorylation event). In order to avoid double counting, multisite phosphorylations are treated as only one data point. Additionally, to test the robustness of our results with respect to the choice of metric, we repeated our analysis on a random subset of ~ 30% of phosphosites using the Local Distance Difference Test, a superposition-free metric of structural similarity[47], using peppr v0.3.1.

The above described comparison is done both for whole structures and for the extracted Pfam domains. In the latter case, we exclude comparisons involving protein kinase C terminal domain (PF00433) for clarity, owing to the very small size of the domain matches compared to the whole protein.

In order to assess whether there is a bias caused by the comparison of structures produced by the same research group, we mined the AUTHOR field from the PDB file headers. Two structures were deemed to be produced by the same research group if they share the same last author. We then performed the comparison described above, but excluding all comparisons between structures sharing the same last author.

Another possible confounding factor is differences in PPI influencing the observed backbone conformational changes between phosphorylated and non-phosphorylated proteins. To investigate this, we analyzed the presence of protein binding partners in phosphorylated and non-phosphorylated structures. For each structure, we deemed a protein chain to be interacting with another within the same structure by computing the buried surface area (BSA) in the complex

with respect to the isolated chains using FreeSASA v2.1.2[117]; an interaction was defined as BSA ≥ 250 Å²[118]. For each phosphosite, we recorded the set of unique binding partners (according to their UniProt ID) in phosphorylated and non-phosphorylated structures, as well as the number of shared partners between them. To assess whether binding interactions could explain backbone conformational changes, we fit a robust linear regression model (to mitigate the influence of outliers) using the median RMSD between phosphorylated and non-phosphorylated structures as the response variable using statsmodels v0.13.5[116]. Predictors included the number of unique partners in phosphorylated structures, the number of unique partners in non-phosphorylated structures, and the number of shared partners between the two. We evaluated the goodness of fit of the model using the pseudo-$R^2$.

To assess whether predicted phosphosite functionality[26] is linked to the degree of conformational change, we performed a two-tailed Mann–Whitney $U$ test. This test compared the distributions of conformational changes between phosphosites predicted to be functional (functional score ≥ 0.5, $n = 60$) and those predicted to be non-functional (functional score < 0.5, $n = 11$). Multisite phosphorylations were excluded from this analysis. Note that functional scores are only available for a subset of phosphosites in our dataset.

To evaluate whether phosphorylation of specific residues might be associated with greater conformational changes, we performed a Kruskal–Wallis test of the median RMSD between phosphorylated and non-phosphorylated structures according to the residue (116 Ser phosphosites, 85 Thr phosphosites, 57 Tyr phosphosites, 24 His phosphosites). Post-hoc analysis was carried out with Dunn's test using FSA v0.9.5[119], and multiple hypothesis testing correction was applied using the Benjamini–Hochberg procedure[115] (FDR < 0.05). Due to small sample sizes, Nd1-phosphonohistidine (HIP) and N1-phosphonohistidine (NEP) are treated indistinctly as phosphorylated histidine. Multisite phosphorylations are excluded from this analysis, since their effect cannot be attributed to a single residue.

## Structural embedding of Pfam domain structures

In order to create a low-dimensional representation of the Pfam domain structures, we used Geometricus v0.1.2[120], which creates protein structure embeddings based on so-called shapemers. We created an embedding for each Pfam domain by fragmenting the structures into shapemers, both based on overlapping k-mers in the sequence ($k = 20$) and on overlapping spheres surrounding each residue (radius = 15 Å). In order to capture fine-grained differences between homologous proteins, a high resolution of 2 was used to discretize rotation invariant moments into shapemers. In order to ensure sufficient sampling and to be able to observe clear clustering of structures, we only embed the structures of a given domain if we have at least 10 structures in total, of which at least 4 must be phosphorylated. 28 different Pfam domains met this criterion. We subsequently reduced the dimensionality of each embedding using PCA.

Finally, we defined clusters of structures using HDBSCAN v0.8.29[121] on the first two principal components using Euclidean distance as a metric. We defined a phosphorylated structure as being within a cluster with non-phosphorylated proteins if there is at least one non-phosphorylated structure within the cluster it belongs to. A structure is defined as not clustered simply if it is labeled by HDBSCAN as not belonging to a cluster. We found no clustering whatsoever in 3 domains with relatively few structures (PF05202, PF13607, PF07804; $n = 11$, 24, and 38, respectively), which we excluded from the analysis.

To evaluate the robustness of our conclusions regarding the structural similarity between phosphorylated and non-phosphorylated protein conformations, we performed clustering directly on the high-dimensional domain embeddings without applying dimensionality reduction. We applied hierarchical clustering as implemented in SciPy v1.9.3[122] using cosine distance and average linkage. The number of

structural clusters for each domain was selected by choosing the number of clusters that maximizes the silhouette score, a metric that quantifies intra-cluster cohesion and inter-cluster separation.

## Ensemble normal mode analysis with atomistic elastic network models

In order to study changes in intrinsic protein dynamics associated with phosphorylation, we performed eNMA with atomistic ENMs) for each set of structures corresponding to each phosphorylation event using the `aanma` function in bio3d v2.4-4[123]. Normal mode analysis (NMA) is a technique grounded in the physics of small oscillations that is used to characterize the flexible states available to a protein near its equilibrium state and provides as output a set of eigenvectors and eigenvalues that describe the directions and amplitudes of motion within the harmonic approximation. Ensemble NMA applies NMA to all homologous protein structures under study. This facilitates the comparison of intrinsic dynamics across homologous but heterogeneous structures (i.e., with unequal sequence composition and length)[69]. Note that while NMA cannot capture large conformational changes or rare events that occur on longer timescales, our ensemble analysis incorporates at once both phosphorylated and non-phosphorylated structures, allowing us to compare their dynamics on a shorter timescale.

While NMA is computationally cheaper than MD simulations, it can still be rather expensive for proteins with thousands of atoms. For this reason, different coarse-grained approximations have been developed, of which the ENM is the most widely used. ENMs are simplified representations of proteins where the protein structure is modeled as a network of nodes and springs connecting nodes within a certain cutoff distance. Here, each structure was represented as an atomistic ENM. Unlike conventional ENMs, where the nodes are defined only by $\alpha$-carbons, atomistic ENMs include other heavy atoms, and have superior predictive performance over classic ENM models[69]. In order to reduce computation time, we used two different approximations. In the rotation-translation block approximation, each residue is assumed to be a rigid body that has only rotational and translational degrees of freedom (i.e., intra-residue deformation is ignored)[124,125]. Furthermore, we also used a coarse-grained representation of each residue. with three to five heavy atoms per residue used to build the ENM. The N, C$\alpha$, and C atoms represent the protein backbone, and zero to two selected side chain atoms represent the side chain (selected based on side chain size and the distance to C$\alpha$). Our analysis otherwise uses the default parameters. Note that while NMA and MD simulations tend to be in broad agreement, the coarse-grained harmonic approach of NMA does not reach the accuracy expected of MD simulations. This is, nonetheless, significantly ameliorated by using the ensemble atomistic approach used here[69].

Residues lacking alpha carbon coordinates were excluded when constructing the networks. Structures with breaks in connectivity (identified using the bio3d `inspect.connectivity` function with default values) due to missing residues were excluded to prevent model bias. We observed no bias towards filtering out phosphorylated or non-phosphorylated structures (Supplementary Fig. 12). Although homology modeling could fill in missing residues, as done in previous work[69], it may also introduce biases such as an increased, artificial homogeneity. Due to these additional filtering steps, we were able to perform this analysis for 111 proteins comprising 128 different phosphorylation events, with 112 being single-site phosphorylations. Finally, in order to reduce memory requirements while preserving as much information as possible, we filtered out highly similar structures using an RMSD cutoff of 0.2 Å. This filtering step is done considering phosphorylated and non-phosphorylated structures separately.

## Global comparison of residue-wise fluctuations

One of the outputs from NMA is predicted residue-wise fluctuations (i.e., variance of atomic positions). This is expressed in squared units (Å²), as variance, by definition, is in squared units. We compared similarities in residue-wise fluctuation profiles by computing their Pearson correlations in the same fashion as with the backbone comparison described above (between all structures phosphorylated at a specific site, between all non-phosphorylated structures and between all phosphorylated and non-phosphorylated structures).

Additionally, we assessed whether phosphorylation is associated with global changes in structural flexibility. For each phosphosite, we computed the median fluctuation for each residue across phosphorylated and non-phosphorylated structures. Differences between the two distributions for each phosphosite ($n = 128$) were assessed using two-tailed Kolmogorov–Smirnov tests, as implemented in SciPy v1.9.3[122]. $P$-values underwent adjustment for multiple hypothesis testing via the Benjamini–Hochberg procedure (FDR < 0.05).

## Local comparison of residue-wise fluctuations

We conducted a comparative analysis of per-residue fluctuations between phosphorylated and non-phosphorylated states to systematically assess the impact of phosphorylation on local dynamics on a case-by-case basis. In order to set a reliable cutoff value for significance, we first evaluated the intrinsic variability present in the dataset. To do so, we computed the absolute value of the per-residue fluctuation differences for all possible pairwise comparisons between phosphorylated and non-phosphorylated proteins. We identified all peaks (i.e., local maxima of absolute differences) and calculated their heights. Outliers were defined as having a height 1.5 times greater or lower than the IQR (the difference between the 25th and the 75th percentiles) and excluded from the dataset. We then deemed a per-residue fluctuation absolute difference between two structures to be significant if it exceeds the median plus 6 standard deviations. This results in a cutoff value of 0.553 Å², representing the top 9.8% of all peaks detected in all pairwise comparisons. As an additional layer, we tested per phosphosite whether the paired per-residue fluctuations come from the same distribution using two-tailed Wilcoxon signed-rank tests ($n = 128$). $P$-values were corrected using the Benjamini–Hochberg method (FDR < 0.05).

Furthermore, we evaluated whether phosphorylation is associated with differential changes in fluctuations around the phosphosite compared to the rest of the protein. We considered any residue whose C$\beta$ (or, in the case of glycine, C$\alpha$) is within 12 Å of the phosphorylated residue's C$\beta$ in any of the relevant structures to be in the neighborhood of the phosphosite. In a few cases, the phosphosite is missing from the structure in one of the groups; such cases are excluded from the analysis, since there is no available reference point. All other protein residues were considered to be part of the background. We then computed the median fluctuations differences between phosphorylated and non-phosphorylated structures for each residue. Differences between the distributions for residues neighboring the phosphosite and the background were evaluated for each phosphosite ($n = 108$) using two-tailed Kolmogorov–Smirnov tests, followed by Benjamini–Hochberg multiple testing correction (FDR < 0.05). Multisite phosphorylations were excluded from the analysis.

## Normal mode comparison

Normal mode analysis decomposes predicted protein motions into separate normal modes. Each normal mode represents a specific pattern of motion in which all parts of the protein oscillate sinusoidally with the same frequency, and sorted by their relative frequency from slowest to fastest. We evaluated the similarity of normal modes using the root mean squared inner product metric (RMSIP), which quantifies the similarity of the directions of normal modes, ranging from 0 (orthogonal motions) to 1 (motions with identical directionality)[69,126]. To evaluate the conservation of normal modes of motion in each phosphorylation event in our dataset, we calculated RMSIPs for all pairs of non-trivial modes $i$ and $j$ between two structures $A$ and $B$. However,

structural variations can result in differences in the relative frequencies of modes across different structures, thereby requiring the identification of equivalent modes. To address this, we formulated the mode-mode matching problem as a linear assignment problem, following the approach of Zhang et al.[72]. The cost of matching modes $i$ and $j$ is defined as 1 - $RMSIP_{i,j}$. We then applied the Hungarian algorithm[127] to find the optimal set of mode pairs that minimizes the total cost.

We subsequently examined changes in structural dynamics as a function of four different frequency regimes of normal modes: global modes ($k = 1$–3), LF modes ($k = 4$–20), LTIF modes ($k = 21$–60), and high frequency modes ($k > 60$). These windows were derived in previous work from the average evolutionary behaviour of 77 different CATH superfamilies, with global modes being the most highly conserved[72]. We stratified these comparisons according to the median RMSIP over the first 20 non-trivial normal modes between phosphorylated and non-phosphorylated structures, according to the degree of divergence upon phosphorylation. Phosphorylation events were categorized into three bins: RMSIP ≥ 0.90 (most similar, $n = 100$), 0.90 > RMSIP ≥ 0.80 ($n = 18$), and RMSIP < 0.80 (least similar, $n = 10$).

### Linear mutual information comparisons

For each structure whose dynamics were modeled with NMA, we derived a matrix of mutual information values. Each entry $ij$ in this matrix quantifies the degree to which the motion of residues $i$ and $j$ are coupled. Values range from 0 (complete independence between residue motions) to 1 (complete dependence). This allows us to detect dependencies in predicted residue motions regardless of their direction or linearity.

We compared these matrices using the $RV_2$ coefficient[94]. The $RV_2$ coefficient is a measure of similarity between two matrices, ranging between −1 and 1, with the same interpretation as Pearson correlation. It corrects the inflated values produced by the original $RV$ coefficient on high dimensional data[94]. $RV_2$ coefficients were calculated using Hoggorm v0.13.3[128].

In order to detect communities of residues with coupled dynamics, we clustered residues using HDBSCAN[121] with default parameters and the mutual information matrix as input.

### Protein strain analysis

PSA calculates mechanical strain by adapting the formalism of finite strain theory to protein structures. Strain is a well-established physical concept that measures how much an object is locally deformed between a reference state and a deformed state. For each C$\alpha$, we define a local neighbourhood of a fixed radius and select all the C$\alpha$ within this sphere. Then, PSA quantifies how the relative positions between the central atom and its neighbours change from the reference to the deformed state. To perform all mechanical deformations analyses, we used the Python library PSA v1.1. See ref. 83 for a comprehensive description of the theoretical framework.

The data used in this analysis comprised two datasets: the protein structures described in the "Structural data collection and curation" subsection of the "Methods" and annotated protein functional sites mined from UniProt (including the categories active site, binding site, DNA binding site, and site). Protein structures with substantial regions of unmodeled residues were excluded, as these gaps in the structure prevent the calculation of local deformations. After filtering, the final dataset consisted of 139 proteins, comprising 179 different phosphosites and 487 protein functional sites.

For each phosphosite, we used PSA to perform all pairwise comparisons between structures in their phosphorylated and non-phosphorylated states. The specific parameters used were atomic selection based on C$\alpha$ and a local neighbourhood radius of 12 Å. The sets of atoms in the neighbourhoods of reference and deformed states were then intersected to compute the pairwise strain. The resulting analyses yielded a residue-wise deformation estimate, which was summarised using the three eigenvalues of the Cauchy strain tensor, corresponding to the principal stretches ($\lambda_i$, with $i = 1, 2, 3$), sorted from smaller to larger. We used as our metric of deformation for all analyses the largest principal stretch $\lambda_3$, which corresponds to the extensive deformation (in contrast to e.g., the compressive deformation measured by $\lambda_1$) of the local neighbourhood.

To quantify the mean extensive stretch $\langle \lambda_3 \rangle$ in each group (including bootstrap samples) in Fig. 5b, we first calculated the geometric mean of strains for each structure, followed by the arithmetic mean of this value across pairwise comparisons. We used two-tailed Mann–Whitney $U$ tests (unpaired) to compare different groups, while Wilcoxon signed-rank tests (paired) were used to compare data to bootstrap samples. Finally, to assess whether a specific site was significantly strained, we compared the mean extensive stretch to bootstrapped values ($n = 50$), with the criterion $\langle \lambda_3 \rangle^{data} > \frac{1}{n} \sum_i^n \langle \lambda_3 \rangle_i^{bootstrap} + 3\sigma^{bootstrap}$, where $\sigma$ is the standard deviation of the bootstrap samples.

To identify the manifold of highly strained residues, we developed an in-house script to implement the method described in ref. 84, using $\lambda_3$ as the measure of deformation. In brief, we identified the manifold $\mathcal{M}$ as the set of $N$ top-strained residues whose correlation dimension $\nu$ (a method to estimate the dimensionality of a set of coordinates in space[129]) is minimal. Further analyses are focused on the presence or absence of the phosphosite and functional sites in that manifold. We considered the representation $R$ of the phosphosite or functional site $s$ in the manifold $\mathcal{M}$ as the mean frequency of residues of $s$ in $\mathcal{M}$ across pairwise comparisons.

For Fig. 5d, we consider the distance between phosphosite and functional site as the minimal Euclidean distance between C$\alpha$s of the phosphosite and of the set of residues that constitute the functional site (when modelled).

### Reporting summary

Further information on research design is available in the Nature Portfolio Reporting Summary linked to this article.

## Data availability

The data analyzed in this study, comprising the dataset of paired phosphorylated structures and their non-phosphorylated counterparts, including their UniProt and Protein Data Bank identifiers, as well as the corresponding processed protein structures, has been deposited in Zenodo with the https://doi.org/10.5281/zenodo.14217157. The dataset of paired structures can also be found as Supplementary Data 1. Source data are provided with this paper.

## Code availability

Custom source code for this project, including scripts and notebooks used to process and analyze the data, is publicly available at https://github.com/evocellnet/phosphocontrol/under the BSD-3 license, together with instructions for running the code. The specific version of the code associated with this publication is archived in Zenodo and is accessible via https://doi.org/10.5281/zenodo.16610129[130]. The PSA library, used for protein strain analysis, can be found at https://github.com/Sartori-Lab/PSA. We reuse code from Geometricus[120] (https://github.com/TurtleTools/geometricus), which is under the MIT license.

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

## Acknowledgements
We thank Sreenath Nair and Mihaly Varadi for their help in retrieving the dataset of phosphorylated and non-phosphorylated structures from the PDB.

## Author contributions
M.C.M., V.H.M., P.S and P.B. designed research. M.C.M. collected and processed data. M.C.M. and V.H.M. conducted analyses, designed figures, and wrote the manuscript. P.S. and P.B. supervised the project and reviewed the manuscript.

## Funding

## Competing interests
The authors declare no competing interests.
