## [Transparent Peer Review file · Nature Communications]

Global comparative structural analysis of responses to protein phosphorylation

Corresponding Author: Dr Pedro Beltrao

Version 0:

Reviewer comments:

Reviewer #1

(Remarks to the Author)

I liked this paper. It not only provides a comprehensive examination of phosphorylation sites, and carries out obvious analyses, including structural analysis of responses to protein phosphorylation, but especially it also asks highly relevant structural and functional considerations. For example, "we asked whether the structures of phosphorylated proteins occupy specific regions within the structural landscape shared by homologous proteins, and whether these are shared with non-phosphorylated structures," and then "Upon clustering structures according to the low-dimensional representations, we found that phosphorylated structures strongly tend to be within clusters containing at least one non-phosphorylated structures". These lead them to conclude that "Phosphorylated conformations are pre-existing and accessible regardless of phosphorylation", since all conformations pre-exist (PMID: 19841628, and Refs which are cited there, e.g. PMID: 10506280, PMID: 10386868, PMID: 10468538, PMID: 10739242). Their questions, along with others throughout the paper, and analyses provide depth.

I would like however to note another point which would be good to relate to. Phosphorylation sites are largely identified based on short sequence logos, although contextual information is also considered. Thus, identification of phosphosites may also simply reflect a statistical occurrence, lacking functional role. In analogy to 'passenger mutations', such phosphosite were coined 'passenger phosphorylation' to emphasize that the presence of a phosphorylation recognition sequence logo does not necessarily imply function (PMID: 33288731). In that sense too, I liked the authors testing that the sites they consider are functions.

Overall, comprehensive analyses of responses to protein phosphorylation on the structural level.

(Remarks on code availability)

none

Reviewer #2

(Remarks to the Author)

Marrero and coworkers present a comparative structural analysis of the response of proteins to phosphorylation. They considered a curated dataset of proteins for which experimentally-determined structures in both phosphorylated and non-phosphorylated states exist. They analyzed the impact of phosphorylation on structural, dynamic and mechanical properties. They observe that there is less structural variability for the subgroup of phosphorylated proteins, that protein dynamics is locally influenced by phosphorylation and that for a small subset of proteins there is a mechanical coupling between the phosphosite and the functional site. From these data, the authors establish insights into possible allosteric events modulated by phosphorylation.

I find this is an interesting study, which goes in line with the findings of previous studies, but this time with a larger dataset. However, I have doubts concerning several methodological parts. Specifically:

1. How was the RMSD computed within the phosphorylated and non-phosphorylated subsets? Was this a comparison across different proteins or among different conformational states for each protein? if it was across different proteins, I wonder if the sequence length (i.e. comparing very short with very long proteins) or the presence of gaps may have

influenced the results. If it was among different conformational states, was the sample size still sufficient, given that only a minority of proteins displayed multiple conformations in either phospho-state? The number of data points in either dataset in Fig 1b was not mentioned.

2. On the per-residue RMSD of Fig 1e, was the calculation weighted by the abundance of these aminoacids in the used structures? could the observed reduction for Ser not be attributed to more of these phosphorylated residues present in the data sets?

3. Given that the RMSDs were so small (in the range of few Angstrom), the robustness of the structure superposition should be ensured. Singular-value-decomposition seems to be a good choice but a residue-residue pair-wise distance metric may have been more suitable as the prior processing superposition step is not required.

4. On the PCA clustering of the PFAM domains: was not this somehow expected given the high structural similarity of these domains and the large abundance of non-phosphorylated species?

5. Table S1 listing the curated data set was not shown in the SI.

6. Figure 4 and L. 205-206: how can thermal fluctuations be altered without increasing nor decreasing them? Overall, the majority of the cases showed not clear global change.

7. Details of the CG model for the NMA analysis were not included (i.e. elastic constants, distances, number of simulations, simulation lengths, validation, etc). In particular the phosphorylation parameters are very important in this case.

8. On the NMA analysis, how much does the outcome depend on the harmonic assumption? To address the question of global vs. local changes, one could carry out PCA on the generated conformational ensembles and check for global principal (non-necessarily harmonic) components.

(Remarks on code availability)

Reviewer #3

(Remarks to the Author)

The manuscript contains a rigorous and comprehensive investigation of the effects that phosphorylation can have on the structure and dynamics of well-structured proteins. The authors use the structures of proteins in phosphorylated and non-phosphorylated versions available in PDB as a foundation for their study, and use elastic network models and protein strain analysis to obtain a deeper insight into protein dynamics. This toolset more or less includes all the standard techniques one would normally use to study allosteric signaling in protein structure (with the exclusion of an important, but computationally expensive Molecular Dynamics).

Understanding the mechanistic effects of Post-translational modifications is one of the key problems in molecular biology at the moment, and the manuscript could make a good reference for the state-of-the art in our understanding of the problem. Perhaps as expected for such a complex problem, the authors do not observe a transparent unified mechanism, but instead find a number of small, but significant effects (structural stabilization, changes in fluctuations, local strain). Particularly interesting is the demonstration of strong preference for conformational selection as a structural effect of phosphorylation. Although most of these observations would probably be difficult to use for predicting (or engineering) the biologically relevant effect of any phosphorylation event, they could be helpful in narrowing down the spectrum of structural mechanisms potentially leading to such effects.

Comments/questions:

0. Due to how the authors select their dataset (and the general bias one could expect when working with PDB structures), it mostly contains phosphosites localized to well-structured protein domains. However, the majority of known phosphosites are found in (conditionally) disordered protein regions, and many of these are known to be involved in biological function by regulating SLIM-domain type protein interactions (there are around 1000 such structures in PDB, but all such cases obviously will not be present in the author's dataset as usually only the phosphorylated, domain-bound state can be crystallized). While the authors briefly state this limitation in the Discussion, expanding on it in the Introduction (perhaps with some estimates of what fraction of phosphosites could be effecting through the mechanisms discussed in the manuscript) would provide an important context for the authors' work.

1. It would be helpful to show how balanced the dataset is in terms of protein domains/families. How much of the conclusions would hold if the analysis was repeated within individual domains/families (or, perhaps, re-weighted according to structural clusters)? The authors work with pfam-based clusters in section 3.2, but seeing such type of analysis for e.g. section 3.1 would be very useful. The concern here is whether some over-represented classes (e.g. protein kinases) are responsible for most of the observed effects.

2. For an unprepared reader, there seems to be a contradiction between the conclusions of section 3.1 and section 3.3.

According to the former (line 142), "... phosphorylation generally leads to small but stabilizing conformational changes". However, according to the latter (line 205), "... phosphorylation commonly alters local thermal fluctuations, without consistently increasing or decreasing them". Thermal fluctuations are often interpreted as a measure of structural stability, so it would be helpful if the authors could comment on this.

(Remarks on code availability)

Reviewer #4

(Remarks to the Author)

Marrero et al. present a detailed analysis of the structural changes that may occur in proteins that undergo phosphorylation to understand the general effects caused by this modification. To this end, the authors extract proteins structures in their phosphorylated and non-phosphorylated states from the protein data bank and compare them to identify trends in structural differences. Conformational changes identified are in general small, but may induce allosteric effects wider in the structure. Dynamic effects are seen through changes to local residue fluctuations and alterations to predicted proteins modes of motion. While infrequent, modifications are found to impact residue-residue couplings, and phosphorylated structures show greater local strain.

This work covers a wide range of structural analyses on single-site phosphorylated proteins, providing insight on the proportions of structures that undergo such effects, and insights into the effects of the modification. There is a clear improvement on previous issues with protein-specific approaches, and general approaches on much smaller datasets. Overall, statistical analyses appear careful. The authors identify and discuss some of this study limitations, exploring potential biases such as using X-ray crystallography which may limit this work to the more stable structures, PDB biases towards functional phosphosites, and modelling for intrinsically disordered proteins. The exploration of potential biases that might have occurred is a very good thing to do, but some further aspects should have been considered (see details below). While it would be interesting to compare the impact on single phosphosites with multi-phosphosites, it was clear in the paper the reasons for focusing on the single sites within the analysis completed.

Overall, this study provides a comprehensive insight into the range of structural changes resulting from phosphorylation. It identifies key trends within phosphorylated proteins and provides a clear approach for analysing the structural effects of PTMs. The text is clearly written and follows a logical flow. Figures, while helpful, are somewhat "messy" (i.e., extremely busy, with colours at time hard to distinguish, and fonts of many different sizes, of which some extremely small). The following are our main comments.

- L165. "phosphorylated proteins share specific clusters with non-phosphorylated ones". Figure 2c and Figure S4 show that some eigenspaces represent <20% of total variance. It is possible that clustering in this space might not be a good representation of the protein's conformational space.
- Figure 2c. Consider other colour choices, green points are extremely hard to see over blue ones. The font of the legends is extremely small (a 200% zoom is required to be able to read it).
- L544. The observation made in Methods section that NMA does not enable observing conformational changes is extremely pertinent. This is an important limitation that should be expanded upon in the Discussion section. It is important to note that, while MD simulations are substantially more computationally expensive, they enable discovering new conformational states of the protein in its native environment. NMA will capture the dynamics around a single state which, when solved via X-ray crystallography, represents a low-energy conformer in a crystal lattice. While this is often a suitable representative of a protein state when in its native environment, this is not always the case. In this context, the conclusion at L231 that "the role of phosphorylation in modulating protein dynamics [is] primarily through significant changes in local thermal fluctuations rather than global flexibility shifts" might be a bit of a stretch, since NMA cannot identify alternation of height of energy barriers separating different states.
- Following the previous point, we can expect that some of the X-ray structures used in this study might involve phosphorylated proteins solved as part of a multimer or bound to one or more small molecules. Shall the analysis include comparisons of bound and unbound states, it should be clarified how the conformational changes associated with the presence of a binding partner can be separated from those caused by a phosphorylation.
- Figure 4d. In the bottom example ("rewiring"), it seems surprising that the long terminal disordered region appears as highly coupled (coloured in red) in the phosphorylated case, but not in the non-phosphorylated one (coloured in gray). In terms of data representation, the caption does not explain why some amino acids are shown explicitly in some protein renderings.

Minor comments:

- L146. typo ("accessible", with two s)
- L198. "we used a signal processing-inspired approach with a stringent cutoff". This statement reads as vague.
- L202. Unclear why fluctuation are in Angstrom², and not Angstrom
- L260. Reference to figures 4c, d, e. Should only refer to d (c is irrelevant, e does not exist).
- Figure 4b. Labels are extremely small
- L368. "In some cases". It is unclear if there are other known PTMs which support the same findings as phosphorylation, or if this is a prediction.

A suggestion to the authors for their next publication (which does not affect our assessment of this work), is that figures and their captions should be grouped. Separating them forces the reviewer to jump back and forth in the document, which is a bit annoying.

(Remarks on code availability)

The code produced to generate the analyses in this work are publicly available in the form of a range of Python scripts and their usage within Jupyter notebooks. While this can in principle be useful, at the moment the README file only describes dependencies of these scripts, but not what the individual scripts contain and how to use them. This constitutes a sizeable barrier to the potentially interested user.

Reviewer #5

(Remarks to the Author)

(Remarks on code availability)

Version 1:

Reviewer comments:

Reviewer #2

(Remarks to the Author)

The authors have adequately addressed my concerns. The revised manuscript reflects the specific actions, taken by the authors, to address my questions. The manuscript has substantially improved after the revision.

(Remarks on code availability)

Reviewer #3

(Remarks to the Author)

The authors have addressed all the questions I had in the initial review, and I believe the manuscript is in good shape.

(Remarks on code availability)

Reviewer #4

(Remarks to the Author)

We thank the authors for carefully considering our comments. They have dedicated significant effort to addressing our suggestions, including the production of new analyses and figures. We are pleased with the improvements made to the paper, and now support its publication.

(Remarks on code availability)

The code usability has been improved with substantial more documentation in all the individual scripts. While further guidance on the overall usage could be beneficial, the purpose and usage of each script is now understandable.

Reviewer #5

(Remarks to the Author)

(Remarks on code availability)

RESPONSE TO REVIEWERS

Reviewer #1:

I liked this paper. It not only provides a comprehensive examination of phosphorylation sites, and carries out obvious analyses, including structural analysis of responses to protein phosphorylation, but especially it also asks highly relevant structural and functional considerations. For example, "we asked whether the structures of phosphorylated proteins occupy specific regions within the structural landscape shared by homologous proteins, and whether these are shared with non-phosphorylated structures," and then "Upon clustering structures according to the low-dimensional representations, we found that phosphorylated structures strongly tend to be within clusters containing at least one non-phosphorylated structures". These lead them to conclude that "Phosphorylated conformations are pre-existing and accessible regardless of phosphorylation", since all conformations pre-exist (PMID: 19841628, and Refs which are cited there, e.g. PMID: 10506280, PMID: 10386868, PMID: 10468538, PMID: 10739242). Their questions, along with others throughout the paper, and analyses provide depth.

I would like however to note another point which would be good to relate to. Phosphorylation sites are largely identified based on short sequence logos, although contextual information is also considered. Thus, identification of phosphosites may also simply reflect a statistical occurrence, lacking functional role. In analogy to 'passenger mutations', such phosphosite were coined 'passenger phosphorylation' to emphasize that the presence of a phosphorylation recognition sequence logo does not necessarily imply function (PMID: 33288731). In that sense too, I liked the authors testing that the sites they consider are functions.

Overall, comprehensive analyses of responses to protein phosphorylation on the structural level.

We appreciate the reviewer's positive feedback on our work and are grateful for the acknowledgment of the depth of our questions and analyses. We have included some of the suggested references on conformational selection that we had missed.

Reviewer #2:

Marrero and coworkers present a comparative structural analysis of the response of proteins to phosphorylation. They considered a curated dataset of proteins for which experimentally-determined structures in both phosphorylated and non-phosphorylated states exist. They analyzed the impact of phosphorylation on structural, dynamic and mechanical properties. They observe that there is less structural variability for the subgroup of phosphorylated proteins, that protein dynamics is locally influenced by phosphorylation and that for a small subset of proteins there is a mechanical coupling between the phosphosite and the functional site. From these data, the authors establish insights into possible allosteric events modulated by phosphorylation.

I find this is an interesting study, which goes in line with the findings of previous studies, but this time with a larger dataset. However, I have doubts concerning several methodological parts. Specifically:

We appreciate these positive remarks by the reviewer.

1. How was the RMSD computed within the phosphorylated and non-phosphorylated subsets? Was this a comparison across different proteins or among different conformational states for each protein? If it was across different proteins, I wonder if the sequence length (i.e. comparing very short with very long proteins) or the presence of gaps may have influenced the results. If it was among different conformational states, was the sample size still sufficient, given that only a minority of proteins displayed multiple conformations in either phospho-state? The number of data points in either dataset in Fig 1b was not mentioned.

To address the first part of this comment, our analysis does not include comparisons between structures of homologs. All comparisons are done between different experimentally solved structures of the same protein (i.e., they must share the same UniProt ID). We have further clarified this point in section 5.1 and at the beginning of section 3.1.

Regarding the second part, differing length between structures can indeed still be a challenge, even if the analysis is done between structures of the same proteins. In our initial analyses, we observed that constructs of the same protein may vary significantly in length (e.g. both short peptides and full domains), and structural overlap can be minimal or even null (e.g. different domains of the same protein). This is precisely why our data curation process includes steps to ensure a minimum coverage of the full protein sequence and consistency (in terms of sequence overlap) of the structures (section 5.1, Fig. 1a, Fig. S1). For greater clarity, we have now emphasized this in the introduction of the Results section.

Regarding Fig. 1b and 1c, these present median values, and only data points with at least two phosphorylated and two non-phosphorylated structures are included. More specifically, Fig. 1b contains 290 data points for the “phosphorylated vs non-phosphorylated” group, 260 for the “within non-phosphorylated” group and 210 for the “within phosphorylated” group. Fig. 1c includes 197 data points, after removing cases where only one structure was available for either the phosphorylated or the non-phosphorylated state. Note that multisite phosphorylations are treated as a single data point to avoid double counting, as explained in section 5.3. We have now included the sample sizes in this section.

Finally, we acknowledge it is not always possible to assess the statistical significance of the conformational changes taking place upon specific phosphorylation events due to the small sample sizes available for many cases. This is precisely why we focus on broader trends in the dataset, rather than individual cases. The dataset is sufficiently large to detect robust overall trends, as reflected in the high statistical significance of our results.

2. On the per-residue RMSD of Fig 1e, was the calculation weighted by the abundance of these aminoacids in the used structures? could the observed reduction for Ser not be attributed to more of these phosphorylated residues present in the data sets?

We appreciate the concern of the reviewer, but we believe that weighting by sample size is not necessary in this analysis. We use the Kruskal-Wallis test to evaluate differences in RMSD upon phosphorylation between different residues. This is a non-parametric method that inherently accounts for different sample sizes (DOI: 10.1080/01621459.1952.10483441).

Thus, the smaller conformational changes associated with phosphorylated serine residues (compared to threonine and tyrosine residues) cannot simply be attributed to their higher abundance in the dataset. Furthermore, while serine constitutes indeed the largest group, reflecting the fact it is the most frequently phosphorylated residue, this trend does not hold across the dataset. Histidine, which has the smallest sample size, also exhibits the smallest conformational changes, illustrating that sample size alone does not drive the observed results.

3. Given that the RMSDs were so small (in the range of few Angstrom), the robustness of the structure superposition should be ensured. Singular-value-decomposition seems to be a good choice but a residue-residue pair-wise distance metric may have been more suitable as the prior processing superposition step is not required.

As the reviewer points out, global superposition-based metrics can have shortcomings, particularly in cases involving flexible or multi-domain proteins where domain rearrangements can disproportionately influence alignment. In such cases, global rigid-body superposition can be dominated by the largest domain and result in inflated RMSD values. To assess the robustness of our structural comparisons, we repeated the backbone comparison analysis using the Local Distance Difference Test (IDDT; DOI: 10.1093/bioinformatics/btt473), a superposition-free metric specifically designed to address these challenges. Note that higher IDDTs correspond to more similar structures, the opposite of RMSD.

We applied IDDT to a random subset of ~30% of phosphosites and found that it reproduces the same trends observed with RMSD (section 3.1, Fig. S2a). Furthermore, we observed a strong correlation between IDDT scores and RMSD values (Fig. S2b). This shows our results are robust with respect to the choice of similarity metric.

We would like to highlight that this finding recapitulates our initial exploratory analysis performing global comparisons of the different sets using normal mode and strain analysis, which do not depend on superposition. Note that these initial results were not included in the manuscript, as the overall message is redundant with what we already showed in the structural comparison.

We trust that, together with the clarification above on our data curation process, this should address the reviewer's overall concerns on the structural comparison.

4. On the PCA clustering of the PFAM domains: was not this somehow expected given the high structural similarity of these domains and the large abundance of non-phosphorylated species?

We assume the reviewer's concern refers to whether the clustering of phosphorylated structures with non-phosphorylated ones in the PCA could be explained solely by the high structural similarity between Pfam domains or the larger abundance of non-phosphorylated structures. However, we do not believe there is any theoretical reason to expect so; the outcome of the analysis depends on mechanism of action, not mere structural similarity or sample size. If phosphorylation consistently induced conformational changes following an induced fit model (i.e., the phosphorylated conformation being inaccessible in the absence of phosphorylation), phosphorylated structures would form distinct "islands" within the PCA space, regardless of the factors highlighted by the reviewer. Nevertheless, we agree with the reviewer that this is an important point for discussion that we have added to the manuscript.

5. Table S1 listing the curated data set was not shown in the SI.

This comes as a surprise, as we did upload the data. We are unsure as to why it was not available and we will work with the journal to ensure it is available.

Note that the curated data is also available as part of a Zenodo repository (<https://zenodo.org/records/14217158>), as highlighted in the Data Availability section. It should be noted that the supplementary tables are different spreadsheets separate from the pdf file that contains the supplementary figures.

6. Figure 4 and L. 205-206: how can thermal fluctuations be altered without increasing nor decreasing them? Overall, the majority of the cases showed not clear global change.

We apologize for the confusion; we now see the potential for misinterpretation. What we mean is that local thermal fluctuations often differ between phosphorylated and non-phosphorylated forms, but there is no consistent trend in whether phosphorylation increases or decreases these fluctuations (as illustrated in Fig. 3e). We have revised the text to make this point clearer. Note that when examining global changes in flexibility, we observe the same lack of consistent directionality (see Fig. S5b and S5c).

7. Details of the CG model for the NMA analysis were not included (i.e. elastic constants, distances, number of simulations, simulation lengths, validation, etc). In particular the phosphorylation parameters are very important in this case.

Unless otherwise stated, our analysis uses the default parameters for the `aanma` function in the R package `bio3d` 2.4-4. Any deviations from these parameters (use of the rotation-translation block approximation and coarse-grained representation of residues by selection of heavy atoms) are noted explicitly; we have made this more clear in section 5.5.

In response to this comment, as well as the following, we would like to clarify some key aspects of NMA analysis:

- Unlike molecular dynamics, NMA does not involve a time-dependent simulation. Instead, the results are derived in a purely analytical manner. There is thus no concept of simulation length in NMA analysis.
- The reviewer inquires about the number of simulations performed. Unlike molecular dynamics, which can incorporate stochasticity due to e.g. random initial velocities, NMA is entirely deterministic. Therefore, there is no need to perform multiple replicates of the analysis. Instead, whenever an ensemble of structures is available, we incorporate this variability directly into our analysis by performing ensemble NMA, which applies NMA to all homologous structures under study at once.
- It is unclear which type of validation the reviewer is requesting. As referenced in the manuscript, NMA has been extensively validated over several decades through comparison against experimental data and molecular dynamics simulations (DOIs: [10.1016/j.str.2007.03.013](https://doi.org/10.1016/j.str.2007.03.013), [10.1002/prot.22855](https://doi.org/10.1002/prot.22855), [10.1146/annurev.biophys.093008.131258](https://doi.org/10.1146/annurev.biophys.093008.131258), [10.1021/acs.jpcb.6b01991](https://doi.org/10.1021/acs.jpcb.6b01991)). This includes the atomistic ensemble NMA approach used in our manuscript, which outperforms other NMA approaches ([10.1021/acs.jpcb.6b01991](https://doi.org/10.1021/acs.jpcb.6b01991)).
- Unlike MD, NMA does not take into account the physicochemical characteristics of the residues such as charge, but the topology of contacts in the structure. Our approach considers phosphorylation implicitly through its effects on the input structural model. Changes in the structural arrangement (and thereby dynamics) due to phosphorylation are reflected in the geometry of the coarse-grained elastic network model.

We hope this explanation provides sufficient clarity regarding the setup of our NMA approach. If additional details are required, we are happy to provide further specifics and adjust the manuscript accordingly.

8. On the NMA analysis, how much does the outcome depend on the harmonic assumption? To address the question of global vs. local changes, one could carry out PCA on the generated conformational ensembles and check for global principal (non-necessarily harmonic) components.

We would like to further clarify the scope and limitations of the NMA analysis:

- Regarding the first point, conventional NMA (including the approach used in our work) is based entirely on a harmonic potential approximation. As such, it is inherently unable to capture anharmonic effects that MD simulations can capture.

This matter was subject of some work, largely in the 1990s, and is reviewed by Atilgan *et al.*, Biophysical Journal (2001) (DOI: [10.1016/S0006-3495\(01\)76033-X](https://doi.org/10.1016/S0006-3495(01)76033-X)).

- Second, regarding the suggestion to carry out PCA on conformational ensembles, it is important to clarify that conventional NMA does not generate conformational ensembles as MD simulations do. It cannot capture large conformational changes or rare events that can otherwise be observed on long-running MD simulations. What NMA does provide is a set of eigenvectors and eigenvalues that describe the directions and amplitudes of motion within the harmonic approximation. The results describe the intrinsic dynamics near the equilibrium state (as described in section 5.5) but are not analogous to an ensemble-based representation of conformational variability. We have further clarified what the output of NMA is in section 5.5.

Hybrid methods that combine NMA with short MD simulations to generate conformational ensembles do exist (e.g., DOI: [10.3389/fmolb.2022.832847](https://doi.org/10.3389/fmolb.2022.832847)). However, implementing such methods at the scale of our dataset is computationally prohibitive. Additionally, while there is some (in practice, seldom used) work incorporating anharmonic behavior into NMA (e.g., DOI: [10.1016/j.bpj.2010.03.027](https://doi.org/10.1016/j.bpj.2010.03.027)), merging these approaches would require significant methodological development, which would merit its own dedicated study and is thus beyond the scope of this work. Besides this, it is unclear to us what additional insights would be gained by incorporating the suggested analysis that are not already covered by the results described in our manuscript.

We hope this explanation clarifies the reviewer's concerns and aligns expectations regarding the capabilities, aims and limitations of NMA as used in our work.

Reviewer #3:

The manuscript contains a rigorous and comprehensive investigation of the effects that phosphorylation can have on the structure and dynamics of well-structured proteins. The authors use the structures of proteins in phosphorylated and non-phosphorylated versions available in PDB as a foundation for their study, and use elastic network models and protein strain analysis to obtain a deeper insight into protein dynamics. This toolset more or less includes all the standard techniques one would normally use to study allosteric signaling in protein structure (with the exclusion of an important, but computationally expensive Molecular Dynamics).

Understanding the mechanistic effects of Post-translational modifications is one of the key problems in molecular biology at the moment, and the manuscript could make a good reference for the state-of-the art in our understanding of the problem. Perhaps as expected for such a complex problem, the authors do not observe a transparent unified mechanism, but instead find a number of small, but significant effects (structural stabilization, changes in fluctuations, local strain). Particularly interesting is the demonstration of strong preference for conformational selection as a structural effect of phosphorylation. Although most of these observations would probably be difficult to use for predicting (or engineering) the biologically relevant effect of any phosphorylation event, they could be helpful in narrowing down the spectrum of structural mechanisms potentially leading to such effects.

We thank the reviewer for their insightful and positive feedback. We agree that, on their own, our current observations are insufficient for directly predicting the effects of specific phosphorylation events. However, we believe that extending this work to incorporate additional factors, such as the local physicochemical environments of phosphosites and relating them to our observations, could enable the identification of major groups of phosphosites that operate through similar structural and dynamic mechanisms. This 'reverse engineering' would, in turn, provide valuable insights for the rational design of novel phosphosites and for understanding their biological effects.

Comments/questions:

0. Due to how the authors select their dataset (and the general bias one could expect when working with PDB structures), it mostly contains phosphosites localized to well-structured protein domains. However, the majority of known phosphosites are found in (conditionally) disordered protein regions, and many of these are known to be involved in biological function by regulating SLIM-domain type protein interactions (there are around 1000 such structures in PDB, but all such cases obviously will not be present in the author's dataset as usually only the phosphorylated, domain-bound state can be crystallized). While the authors briefly state this limitation in the Discussion, expanding on it in the Introduction (perhaps with some estimates of what fraction of phosphosites could be effecting through the mechanisms discussed in the manuscript) would provide an important context for the authors' work.

The reviewer raises an important point; the bias towards well-structured proteins in datasets derived from the PDB is a limitation that ought to be clearly acknowledged and contextualized. In a prior study (DOI: 10.1038/s41467-019-09952-x) examining the conservation of phosphorylation within protein domain families, our lab compiled data from 40 eukaryotic species, identifying 537,321 unique phosphosites. Of these, 83,359 (approximately 15.5%) were mapped directly to Pfam domain regions. This figure likely represents a conservative estimate of phosphosite localization within structured regions in eukaryotes, as it excludes sites adjacent to domains, which likely operate through similar mechanisms. It should be considered that phosphosites located in protein domains are more likely to play functional roles (DOI: 10.1038/s41587-019-0344-3). We have expanded on this point in the introduction.

1. It would be helpful to show how balanced the dataset is in terms of protein domains/families. How much of the conclusions would hold if the analysis was repeated within individual domains/families (or, perhaps, re-weighted according to structural clusters)? The authors work with pfam-based clusters in section 3.2, but seeing such type of analysis for e.g. section 3.1 would be very useful. The concern here is whether some over-represented classes (e.g. protein kinases) are responsible for most of the observed effects.

We thank the reviewer for highlighting the importance of assessing potential biases due to uneven representation of protein families. There is a strong interest in studying protein kinases; this is reflected in the PDB, and therefore, our dataset. 130 out of 347 (37%) phosphosites within our dataset occur within protein kinase domains, with no other domain

coming close to this level of representation (Fig. S4a). This can also be seen at the overall sequence level, rather than just the domain level. By clustering the 225 sequences in the datasets using a 30% overall identity threshold with MMSeqs2, we obtain 160 different clusters, only 10 of which contain more than 2 sequences; half of these clusters contain protein kinase domains (note that, because of different domain architectures, not all protein kinases will be in the same cluster). This shows that the imbalance in the dataset is overwhelmingly due to protein kinases.

To investigate whether this overrepresentation could be driving our overall observations, we stratified the analysis into two groups: phosphosites located within protein kinase domains and those found in any other domains. Our analysis reveals a statistically significant difference in the extent of conformational changes upon phosphorylation between these two groups (Fig. S4b). Specifically, phosphosites within protein kinase domains are associated with a median backbone RMSD change of 1.51 Å, compared to 0.73 Å for phosphosites not in kinase domains (one-sided Mann-Whitney U test, adjusted p-value = 4.92×10^{-11}). For reference, we found a median backbone RMSD change of 1.14 Å over the whole dataset.

Importantly, we also examined whether the trend of larger conformational changes in kinase domains holds independently of phosphorylation. We found that the median backbone RMSD is significantly higher for structures involving protein kinase domains across both phosphorylated and non-phosphorylated sets (Fig. S4b). Specifically, within phosphorylated proteins, kinase domain regions show larger conformational changes than non-kinase regions (one-sided Mann-Whitney U test, adjusted p-value = 2.30×10^{-6}). The same trend is observed within non-phosphorylated proteins (adjusted p-value = 1.13×10^{-6}). This suggests that, in general, conformational changes associated to protein kinase domains are larger than usual, regardless of phosphorylation.

Finally, when we compared the variability of the two distributions, we did not observe significant differences between backbone RMSD upon phosphorylation associated with phosphosites in protein kinase domains versus phosphosites elsewhere (Brown-Forsythe test, p -value = 0.968). This indicates that, while the median magnitude of conformational change is higher in protein kinase domains, phosphorylation does not necessarily lead to similar conformational changes simply because it takes place in the same domain.

To summarize, the dataset is biased towards protein kinases, as pointed out by the reviewer. While this bias may inflate the observed median RMSD across all groups (a trend opposed by what we call the “lead author effect”, which makes our overall comparison more conservative), the core trends we report (small conformational changes upon phosphorylation, increased structural uniformity among phosphorylated structures) remain consistent regardless of the presence or absence of protein kinases.

2. For an unprepared reader, there seems to be a contradiction between the conclusions of section 3.1 and section 3.3. According to the former (line 142), “... phosphorylation generally leads to small but stabilizing conformational changes”. However, according to the latter (line 205), “... phosphorylation commonly alters local thermal fluctuations, without consistently increasing or decreasing them”. Thermal fluctuations are often interpreted as a measure of structural stability, so it would be helpful if the authors could comment on this.

Thank you for pointing out this apparent contradiction. Our description of phosphorylation as “stabilizing” refers to ensemble-level observations: across multiple experimentally solved structures, the phosphorylated forms tend to be more similar to each other than the non-phosphorylated ones are. In other words, phosphorylation appears to constrain the conformational ensemble, which we interpret as a stabilizing effect on the overall structure. In contrast, the changes in thermal fluctuations represent local residue mobility around a given structure, not the spread of a conformational ensemble. These statements describe different things (ensemble uniformity versus local thermal fluctuations) and can coexist. We have revised the text to make this distinction more explicit at the end of sections 3.1 and 3.3.

Reviewer #4

Marrero *et al.* present a detailed analysis of the structural changes that may occur in proteins that undergo phosphorylation to understand the general effects caused by this modification. To this end, the authors extract proteins structures in their phosphorylated and non-phosphorylated states from the protein data bank and compare them to identify trends in structural differences. Conformational changes identified are in general small, but may induce allosteric effects wider in the structure. Dynamic effects are seen through changes to local residue fluctuations and alterations to predicted proteins modes of motion. While infrequent, modifications are found to impact residue-residue couplings, and phosphorylated structures show greater local strain.

This work covers a wide range of structural analyses on single-site phosphorylated proteins, providing insight on the proportions of structures that undergo such effects, and insights into the effects of the modification. There is a clear improvement on previous issues with

protein-specific approaches, and general approaches on much smaller datasets. Overall, statistical analyses appear careful. The authors identify and discuss some of this study limitations, exploring potential biases such as using X-ray crystallography which may limit this work to the more stable structures, PDB biases towards functional phosphosites, and modelling for intrinsically disordered proteins. The exploration of potential biases that might have occurred is a very good thing to do, but some further aspects should have been considered (see details below). While it would be interesting to compare the impact on single phosphosites with multi-phosphosites, it was clear in the paper the reasons for focusing on the single sites within the analysis completed.

Overall, this study provides a comprehensive insight into the range of structural changes resulting from phosphorylation. It identifies key trends within phosphorylated proteins and provides a clear approach for analysing the structural effects of PTMs. The text is clearly written and follows a logical flow. Figures, while helpful, are somewhat “messy” (i.e., extremely busy, with colours at time hard to distinguish, and fonts of many different sizes, of which some extremely small). The following are our main comments.

We thank the reviewer for the positive feedback. We would like to clarify that our analysis does include multi-site phosphorylation where possible; however, the dataset does primarily consist of single-site phosphorylation events (79% of phosphosites). We have further clarified this at the beginning of the results section.

- L165. "phosphorylated proteins share specific clusters with non-phosphorylated ones". Figure 2c and Figure S4 show that some eigenspaces represent <20% of total variance. It is possible that clustering in this space might not be a good representation of the protein's conformational space.

We appreciate the reviewer's concern regarding the low variance explained by the first two principal components in some Pfam domains. To address this, we performed hierarchical agglomerative clustering directly on the high-dimensional domain embeddings, without PCA, to verify if we reach the same conclusion. We chose hierarchical clustering in this case over the HDBSCAN method used in the manuscript because density-based clustering algorithms often perform poorly in high-dimensional spaces. Note that hierarchical clustering also differs from HDBSCAN in that it assigns all samples to clusters.

We used cosine distance with average linkage and selected the number of clusters using the silhouette score, a measure of how well samples are clustered. Repeating our original analysis (measuring the fraction of phosphorylated structures clustered with non-phosphorylated ones per Pfam domain) we found very similar results (Fig. S7). This supports our conclusion that phosphorylated conformations are pre-existing and accessible without the need for phosphorylation. The fact that in some cases there are low explained variance ratios in the first two principal components, following the approach in the manuscript, does not detract from this conclusion.

We believe the observation that some domains exhibit <20% of their total variance in the first two principal components is more easily attributable to characteristics of each particular set of domain structures. For example, in domains where structural diversity is distributed more evenly across orthogonal axes, no single principal component would dominate the variance. We observe, for example, as we show in the following scatterplot, a clear correlation (Pearson $r = 0.48$) between the cumulative variance explained by the first two principal components and the median cosine similarity of the shapemer count embeddings across Pfam domain families. This indicates that sets of domain structures that are overall more homogeneous have variance that is more easily explained by PCA.

- Figure 2c. Consider other colour choices, green points are extremely hard to see over blue ones. The font of the legends is extremely small (a 200% zoom is required to be able to read it).

We have replaced green with dark orange for greater contrast, as well as increased the font size of the legends.

- L544. The observation made in Methods section that NMA does not enable observing conformational changes is extremely pertinent. This is an important limitation that should be expanded upon in the Discussion section. It is important to note that, while MD simulations are substantially more computationally expensive, they enable discovering new conformational states of the protein in its native environment. NMA will capture the dynamics around a single state which, when solved via X-ray crystallography, represents a low-energy conformer in a crystal lattice. While this is often a suitable representative of a protein state when in its native environment, this is not always the case. In this context, the conclusion at L231 that “the role of phosphorylation in modulating protein dynamics [is] primarily through significant changes in local thermal fluctuations rather than global flexibility shifts” might be a bit of a stretch, since NMA cannot identify alternation of height of energy barriers separating different states.

We fully agree that NMA, by design, captures dynamics around a single conformational minimum and is inherently unable to reveal large-scale transitions or alternate conformational states that may be functionally relevant. We wish to highlight, however, that we mitigate this limitation by analyzing (where available) ensembles of structures for both the phosphorylated and non-phosphorylated states (see section 5.5). In this way, we can capture a broader range of conformational states and compare directly the dynamic behaviour of ensembles. For greater clarity, we have further emphasized this aspect in section 3.3.

Regarding the sentence in question, we agree that the original phrasing may have overstated the conclusion. Our intention was to highlight the observed differences in local residue fluctuations, rather than to make broader claims about energy landscapes or transitions between conformational states. We have therefore revised the sentence for greater precision. We also wish to remark that the analyses shown before in our manuscript do point towards phosphorylation leading to greater structural uniformity due to conformational selection, suggesting indeed changes in the energy barriers.

- Following the previous point, we can expect that some of the X-ray structures used in this study might involve phosphorylated proteins solved as part of a multimer or bound to one or more small molecules. Shall the analysis include comparisons of bound and unbound states, it should be clarified how the conformational changes associated with the presence of a binding partner can be separated from those caused by a phosphorylation.

The reviewer raises an important point. Prior studies have examined factors such as crystallization conditions or crystal contacts, or among the factors the reviewer mentions, presence of small molecules, in relation to phosphorylation-induced conformational changes (DOI: 10.1093/bioinformatics/bts541), finding that such factors do not explain them. However, to our knowledge, no work has specifically disentangled the effects of complex formation.

a)

	Coefficient	p-value
Intercept	1.42	<0.001
Number of unique protein partners (phosphorylated)	0.18	<0.001
Number of unique protein partners (non-phosphorylated)	0.01	0.442
Number of shared protein partners	-0.14	0.05

b)

To address this, we analyzed whether the presence of additional protein binding partners could influence the observed RMSD between phosphorylated and non-phosphorylated structures. For each structure in our dataset, we defined a protein chain as interacting with another if the buried surface area in the complex is at least 250 Å² relative to the isolated chains.

We then assessed to what extent the observed backbone conformational changes might be explained by differences in binding interactions. Specifically, we fit a robust linear model (to downweight the influence of RMSD outliers) using, for each phosphosite, the following predictors: the number of unique binding partners in phosphorylated structures, the number of unique partners in non-phosphorylated structures, and the number of shared partners between both. We found that the number of unique binding partners in phosphorylated structures had a positive small but statistically significant effect on RMSD (model coefficient: 0.18, $p < 0.001$). Likewise, the number of shared partners has a small negative but statistically significant effect on RMSD (model coefficient: -0.14, $p = 0.05$) (Fig. S5a). However, the model's fit was extremely poor (pseudo $R^2 = -0.028$), indicating that the model fits slightly worse than simply predicting the mean, and thus cannot explain the observed conformational changes (Fig. S5b).

In summary, while there appears to be a minor association between changes in binding partners and conformational change, protein-protein interactions overall cannot explain the phosphorylation-dependent structural differences we observe in our dataset.

• Figure 4d. In the bottom example (“rewiring”), it seems surprising that the long terminal disordered region appears as highly coupled (coloured in red) in the phosphorylated case, but not in the non-phosphorylated one (coloured in gray). In terms of data representation,

the caption does not explain why some amino acids are shown explicitly in some protein renderings.

The outcomes of normal mode analysis are sensitive to the distances between residues. Upon phosphorylation, the distances between phosphosites and neighboring residues are altered, as evidenced by the following heatmap. Many residues surrounding the phosphosites are charged, either positively or negatively, which likely contributes to the observed changes in residue-residue distances and thereby couplings. Therefore, the pronounced changes in residue-residue couplings following phosphorylation are not entirely unexpected. We have edited the manuscript to clarify this and added a supplementary figure illustrating the changes in residue-residue distances upon phosphorylation (Fig. S13).

Regarding the second part of the comment, we thank the reviewer for pointing out the omission in the figure caption. Residues shown as sticks indicate phosphosites; we have now clarified this in both the figure caption and the main text. While it may not be immediately obvious, phosphosites are actually rendered explicitly in all protein structures. In the non-phosphorylated form of Ca^{2+} /CaM-dependent protein kinase type II (2bdw_B), the relevant residue is part of the long C-terminal α -helix and is within the pink-coloured cluster. In the phosphorylated form, this region undergoes an order-to-disorder transition and is part of the brown-coloured cluster. In the non-phosphorylated form, the phosphosite is buried between two helices, making it challenging to clearly display both the mutual information clusters and the residue itself; we prioritized the former in the visual representation. We have nonetheless clarified the location of the phosphosite in the caption.

Minor comments:

- L146. typo ("accessible", with two s)

Thank you for spotting this, we have fixed the two occurrences of this typo.

- L198. “we used a signal processing-inspired approach with a stringent cutoff”. This statement reads as vague.

We have modified this statement to make it more clear and specific.

- L202. Unclear why fluctuation are in Angstrom², and not Angstrom

The bio3d package computes residue fluctuations as the variance of atomic positions. Variance, by definition, is in squared units (in this case, Å²). This approach also mirrors the reporting of B-factors in protein crystallography, where B-factors represent the mean squared displacements of atoms due to thermal motion and are likewise expressed in Å². We have clarified this in the relevant methods section (section 5.6).

- L260. Reference to figures 4c, d, e. Should only refer to d (c is irrelevant, e does not exist).

This was a vestige of a previous version; it has now been corrected.

- Figure 4b. Labels are extremely small

Good point; we have increased the font size of the labels, and the figure should be much more legible now.

- L368. “In some cases”. It is unclear if there are other known PTMs which support the same findings as phosphorylation, or if this is a prediction.

This statement is a prediction based on the broad applicability of the conformational selection framework to various molecular interactions, including protein-ligand, protein-protein, and protein-nucleic acid interactions. Consequently, we hypothesize that our findings may well extend to other PTMs as well. While conformational selection driven by PTMs beyond phosphorylation has been suggested in the literature (DOI: [10.1093/bioinformatics/bts541](https://doi.org/10.1093/bioinformatics/bts541)), we are not aware of specific experimental demonstrations of this phenomenon for other PTMs. Note that there are cases where we do not expect conformational selection to necessarily play a role; for example, direct modification of an enzymatic active site. We have revised the text to clarify this point.

A suggestion to the authors for their next publication (which does not affect our assessment of this work), is that figures and their captions should be grouped. Separating them forces the reviewer to jump back and forth in the document, which is a bit annoying.

Grouping figures with their captions is also our strong preference (and likely that of most readers and researchers). We have done so in the revised version of the manuscript.

Reviewer #4 (Remarks on code availability):

The code produced to generate the analyses in this work are publicly available in the form of a range of Python scripts and their usage within Jupyter notebooks. While this can in principle be useful, at the moment the README file only describes dependencies of these scripts, but not what the individual scripts contain and how to use them. This constitutes a sizeable barrier to the potentially interested user.

We appreciate the reviewer's suggestion and agree that clearer guidance can enhance usability. While we believe, however, that exhaustively documenting each and every script and notebook in the README would be somewhat impractical and potentially overwhelming, we have improved the overall clarity and accessibility of the code by expanding documentation, particularly through more detailed docstrings within the scripts themselves.

Reviewer #5

We thank the reviewer for their time evaluating the manuscript.